# miR-486 is essential for muscle function and suppresses a dystrophic transcriptome

Adrienne Samani[1],†, Rylie M Hightower[1,2],†, Andrea L Reid[1], Katherine G English[1] , Michael A Lopez[1,2] , J Scott Doyle[3], Michael J Conklin[3], David A Schneider[4] , Marcas M Bamman[2], Jeffrey J Widrick[5] , David K Crossman[6] , Min Xie[7], David Jee[8,9], Eric C Lai[8,9] , Matthew S Alexander[1,2,6,10]

**miR-486 is a muscle-enriched microRNA, or "myomiR," that has reduced expression correlated with Duchenne muscular dystrophy (DMD). To determine the function of miR-486 in normal and dystrophin-deficient muscles and elucidate miR-486 target transcripts in skeletal muscle, we characterized *mir-486* knockout mice (*mir-486* KO). *mir-486* KO mice developed disrupted myofiber architecture, decreased myofiber size, decreased locomotor activity, increased cardiac fibrosis, and metabolic defects were exacerbated in *mir-486* KO:*mdx^{5cv}* (DKO) mice. To identify direct in vivo miR-486 muscle target transcripts, we integrated RNA sequencing and chimeric miRNA eCLIP sequencing to identify key transcripts and pathways that contribute towards *mir-486* KO and dystrophic disease pathologies. These targets included known and novel muscle metabolic and dystrophic structural remodeling factors of muscle and skeletal muscle contractile transcript targets. Together, our studies identify miR-486 as essential for normal muscle function, a driver of pathological remodeling in dystrophin-deficient muscle, a useful biomarker for dystrophic disease progression, and highlight the use of multiple omic platforms to identify in vivo microRNA target transcripts.**

## Introduction

Skeletal muscle is a remarkable organ with intrinsic plasticity because of its regenerative and reparative capabilities in response to exercise, traumatic injury, and disease. The ability to respond to these stimuli require tunable changes in gene expression often under the control of noncoding RNAs (Chen et al, 2020; Aránega et al, 2021; Archacka et al, 2021; La Rocca et al, 2021). A subset of miRNAs have been identified and shown to be muscle-enriched, play key functional roles in myogenesis and muscle function, and are referred to as myomiRs (Horak et al, 2016). In DMD patients, miRNAs isolated from DMD patient serum are used as biomarkers for disease progression and have been referred to as "dystromiRs" (Zaharieva et al, 2013). When used in combination with a therapeutic treatment such as exon-skipping phosphorodiamidate morpholino oligomers or peptide-conjugated phosphorodiamidate morpholino oligomers, these dystromiRs can be restored to normal levels, thus making them attractive candidates for marking dystrophic disease progression (Coenen-Stass et al, 2018). *mir-486* is a myomiR whose stem loop sequence is embedded within the *ANKYRIN1* (*ANK1*) locus and is strictly conserved across mammals (Small et al, 2010; Alexander et al, 2011). Previously, we demonstrated that miR-486 regulates a DOCK3/PTEN/AKT signaling axis in skeletal muscle, and muscle-specific overexpression of miR-486 improves dystrophic symptoms in dystrophin-deficient (*mdx^{5cv}*) mice (Alexander et al, 2014). Although the reduction of miR-486 expression in dystrophic muscle is known, the suitability of miR-486 as a DMD disease progression biomarker, the mechanism for miR-486 dysregulation in DMD muscle, the in vivo targets of miR-486 in muscle, and the functional consequences of the genetic ablation of *mir-486* in mice in normal and dystrophic muscle remain to be elucidated.

## Results

We first assessed miR-486 levels in normal and DMD patient skeletal muscle biopsies to define its expression with regards to ambulatory status, an important biomarker of DMD disease

[1]Department of Pediatrics, Division of Neurology at Children's of Alabama and the University of Alabama at Birmingham, Birmingham, AL, USA [2]University of Alabama at Birmingham Center for Exercise Medicine (UCEM), Birmingham, AL, USA [3]Department of Orthopedic Surgery, at the University of Alabama at Birmingham, Birmingham, AL, USA [4]Department of Biochemistry and Molecular Genetics at the University of Alabama at Birmingham, Birmingham, AL, USA [5]Division of Genetics and Genomics at Boston Children's Hospital, Boston, MA, USA [6]Department of Genetics, University of Alabama at Birmingham, Birmingham, AL, USA [7]Division of Cardiovascular Disease, Department of Medicine, University of Alabama at Birmingham, School of Medicine, Birmingham, AL, USA [8]Developmental Biology Program, Sloan Kettering Institute, New York, NY, USA [9]Weill Graduate School of Medical Sciences, Cornell University, New York, NY, USA [10]UAB Civitan International Research Center (CIRC), at the University of Alabama at Birmingham, Birmingham, AL, USA

Correspondence: matthewalexander@uabmc.edu
†Adrienne Samani and Rylie M Hightower are co-first authors.

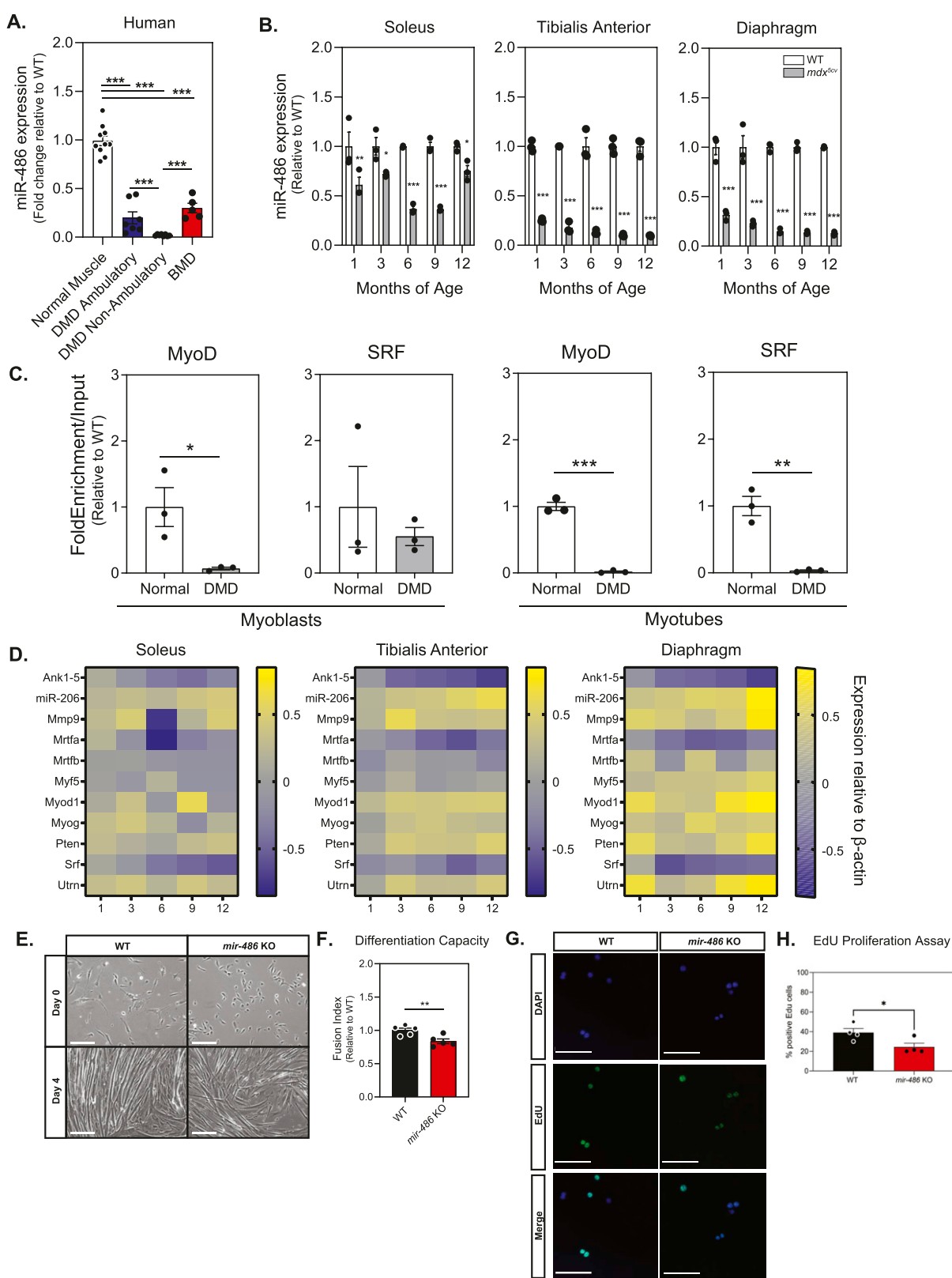

**Figure 1. *miR-486* skeletal muscle expression decreases in DMD.**
**(A)** Quantitative PCR reveals decreased expression of miR-486 in dystrophic human skeletal muscle compared with control human skeletal muscle and Becker muscular dystrophy muscle (N = 8 individual samples: normal muscle, DMD ambulatory, and DMD nonambulatory; N = 5 for Becker muscular dystrophy samples).
**(B)** Quantitative PCR reveals decreasing expression of miR-486 in the *mdx^{5cv}* dystrophic mouse model skeletal muscle at 1, 3, 6, 9, and 12 mo of age compared with WT

progression (Fowler et al, 2018; Rooney et al, 2020). Expression of miR-486 in ambulatory DMD quadriceps biopsies was significantly lower than healthy quadriceps (Fig 1A). miR-486 expression in DMD nonambulatory muscle also decreased compared with both healthy muscle and ambulatory DMD muscle (Fig 1A). Skeletal muscle biopsies from Becker muscular dystrophy patients showed a less decrease in miR-486 expression to DMD, demonstrating that reduced miR-486 is correlated to pathology related to the loss or decrease of a functional dystrophin protein (Fig 1A). To evaluate this in a DMD mouse model, we assessed miR-486 levels in three different muscle groups across five DMD disease-relevant time points (1, 3, 6, 9, and 12-mo-old) in both $mdx^{5cv}$ and wild-type mice. The $mdx^{5cv}$ mice demonstrated significantly decreased miR-486 expression in the tibialis anterior (TA), soleus, and diaphragm compared with WT mice (Fig 1B). This decrease in miR-486 expression remained lower overall in expression when normalized to WT muscle controls at each DMD disease-relevant time point (Fig 1B). MyoD and SRF are important transcriptional activators in skeletal muscle that drive myogenic differentiation of satellite cells into mature myofibers during differentiation and repair (Guerci et al, 2012; Yamamoto et al, 2018). Myogenic differentiation and muscle fiber repair mechanisms are impaired in DMD pathology (Chang et al, 2016; Houang et al, 2018). We sought to determine if MyoD and SRF signaling played a role in the regulation of miR-486 expression by evaluating MyoD- and SRF-binding dynamics within the miR-486 host gene ANKRYIN1 (ANK1), specifically at the muscle-enriched ANK1-5 promoter region. Using chromatin immunoprecipitation (ChIP), both MyoD and SRF DNA-binding at the human ANK1-5 promoter were decreased in DMD patient myoblasts and myotubes (Fig 1C). This result led us to investigate the expression levels of other MyoD- and SRF-related myogenic signaling factors that drive miR-486 expression across the lifespan of $mdx^{5cv}$ mice in affected dystrophic muscles. Quantitative PCR of MyoD and SRF signaling factors revealed decreased expression in correlation with dystrophic disease progression in the $mdx^{5cv}$ mouse muscles (Fig 1D).

As miR-486 expression is significantly and progressively decreased in DMD, we sought to map its functional role in skeletal and cardiac muscle. mir-486 KO mice generated by CRISPR exhibit an 85-bp deletion within the pre-miRNA hairpin (Fig S1A and B). Ablation of mir-486 was further validated via Northern blot as it was undetectable in both mir-486 KO TA and whole heart (Fig S1C). Western immunoblotting demonstrated that ANK1-5 protein expression was unperturbed in the mir-486 KO mice that was further confirmed using quantitative RT-PCR showing no disruption of Ank1-5 transcript levels (Fig S1D and E). These results demonstrated that mir-486 KO-associated muscle pathologies are unlikely to involve disruption of its host gene. We evaluated the role of miR-486 in

myogenic differentiation by measuring myogenic differentiation capacity in the mir-486 KO primary myoblasts. The mir-486 KO myoblasts showed decreased myogenic differentiation capacity compared with WT controls (Fig 1E and F). These results aligned with our previous findings that mir-486 shRNAi knockdown in myoblasts delayed myogenic differentiation and decreased overall differentiation capacity (Alexander et al, 2014). We next wanted to determine what effects the genetic loss of mir-486 had on overall muscle cell proliferation rate and differentiation capacity. We observed a significant decrease in the proliferation rate observed in the mir-486 KO primary myoblasts may be attributed to other defects in muscle hypertrophic growth and defects in cellular differentiation (Fig 1G and H). These findings are consistent with the previously published literature in which muscle myomiRs regulate essential phases of myogenic processes such as cell proliferation, differentiation, muscle function, and myogenic fusion (Chen et al, 2006, 2010; McCarthy & Esser, 2007; Dey et al, 2011; Alexander et al, 2013). The reduction in proliferation in the absence of the miRNA may delegate miR-486 as an essential driver of muscle cell proliferation and myogenic differentiation.

To determine how mir-486 genetic loss affects skeletal muscle, we examined histological sections of control and mir-486 KO TA muscles (Fig 2A). mir-486 KO mice exhibit decreased overall myofiber cross-sectional area, increased centralized myonuclei, and increased fibrosis that was exacerbated in the dystrophic $mdx^{5cv}$ background (Fig 2B–D). We asked if these disruptions played a larger role in whole body composition or locomotion in the mir-486 KO mice. Interestingly, mir-486 ablation did not significantly alter fat mass, lean mass, or overall total body mass (Fig S2A–C). In addition, mir-486 KO did not affect extensor digitorum muscle strength, as measured by peak force per unit physiological cross-sectional area, suggesting that miR-486's function may not directly explain DMD physiological force deficits (Fig S2D). DMD-associated dilated cardiomyopathy is a cardinal hallmark of advanced DMD pathology, and given the high expression of miR-486 in cardiomyocytes, we next examined the functional consequences of mir-486 disruption on heart function (Buddhe et al, 2018). Notably, histopathological analysis of mir-486 KO mouse hearts showed increased fibrosis compared with WT controls (Fig 3A and B). Echocardiograms of the adult mir-486 KO mice revealed a decreased fractional shortening, ejection fraction, and several other parameters that were exacerbated in the mir-486 KO:$mdx^{5cv}$ (DKO) mice (Fig 3C–M). These results demonstrate that miR-486 is involved in pathological remodeling during DMD-associated dilated cardiomyopathy.

Because of enrichment of miR-486 in skeletal muscle, we sought to define the relationship between the loss of miR-486 and the

---

control muscle (N = 3 samples per genotype and time point). **(C)** ChIP revealed myogenic factors MyoD and SRF demonstrate decreased binding at the promoter of ANK1-5 in isolated $mdx^{5cv}$ myoblasts and differentiated myotubes compared with WT controls (n = 3 replicates). **(D)** Heat maps demonstrate changes in expression of myogenic factors in $mdx^{5cv}$ tibialis anterior, soleus, and diaphragm muscles over 1, 3, 6, 9, and 12 mo of age compared with WT control muscle. Yellow indicates an increase in expression and blue indicates a decrease in expression relative to WT control muscle (N = 3 replicates). **(E)** Phase contrast reveals disrupted myoblast differentiation capacity. Photomicrographs show differentiated myotubes after 4 d of culturing from primary isolated satellite cells. WT and mir-486 KO satellite cells were isolated from < p10 pups and cultured for 4 d. 10×, scale bar = 200 μm. **(F)** Myogenic fusion indices calculated from day 4 myotube culture images depicted as calculated by dividing the number of nuclei within multinucleated myofibers by the total number of nuclei (N = 5 replicates per genotype). **(G)** Images assessing myoblast proliferation in mir-486 KO mice versus WT. Cells were stained with DAPI and EdU and quantified. **(H)** Quantification of the EdU proliferation assay, N = 4 separate fields of 100 cells per genotype cohort (mean ± SEM).

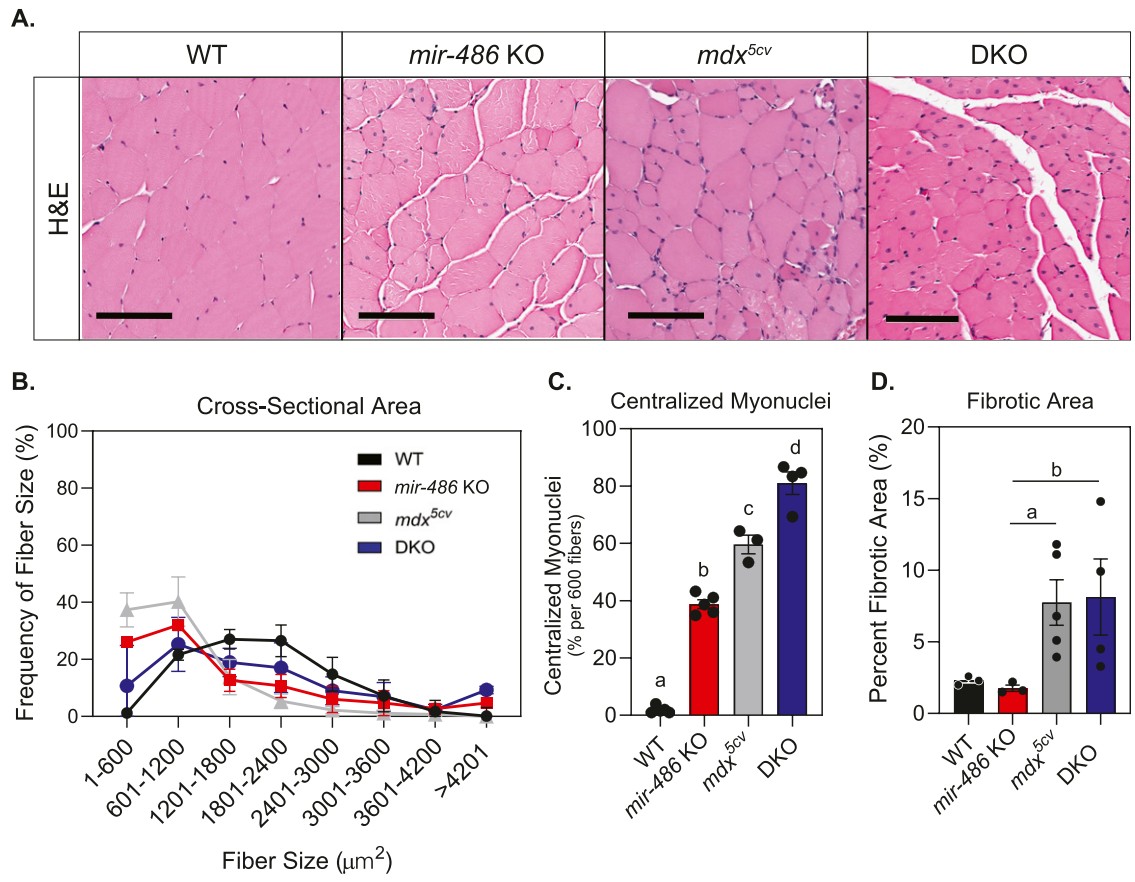

**Figure 2.** ***mir-486* knockout mice demonstrate histological defects in skeletal muscle.**
**(A)** H&E staining of transverse sections of tibialis anterior (TA) muscles at 6 mo of age. Scale bars = 200 $\mu$M. **(B)** Cross-sectional area of myofibers in TA muscles were measured using ImageJ based on H&E staining. Six hundred fibers from five mice of each genotype were counted. **(C)** Centralized myonuclei in WT and *mir-486* KO TA muscle at 6 mo of age were counted using ImageJ. Six hundred fibers from five mice of each genotype were counted. Means with different letters are significantly different (Tukey's HSD, $P < 0.05$). Sections cut at a thickness between 7 and 15 $\mu$m. **(D)** Fibrotic area was quantified as a percentage of total area using ImageJ. Five mice of each genotype were counted. *$P \leq 0.05$. N = 5 mice per genotype cohort.

expression of the four DMD disease-relevant dystromiRs, miR-1, miR-133a, miR-133b, and miR-206. Interestingly, only miR-133b was decreased in *mir-486* KO TA muscles (Fig 4A–D). To understand the effects of *mir-486* ablation on the overall skeletal muscle transcriptome, we performed RNA sequencing on *mir-486* KO and WT TA muscles. RNA sequencing revealed that 96 transcripts significantly increased in expression, and 190 transcripts significantly decreased in expression in *mir-486* KO TA muscles compared with WT (Fig 4E). To identify pathways preferentially targeted by miR-486, we used g: Profiler software to perform pathway enrichment analysis on the 96 transcripts that significantly increased in expression and the 190 transcripts that significantly decreased in expression in the *mir-486* KO TA muscles. The top 5 enriched pathways identified by g:Profiler were extracellular matrix, collagen-containing extracellular matrix, regulation of multicellular organismal processes, extracellular region, and vasculature development as shown by the g:GOSt multi-query Manhattan plot (Fig 4F). The top 10 transcripts increased in expression and top 10 transcripts decreased in expression in *mir-486* KO compared with WT are shown (Fig 4G and H). To evaluate targets of miR-486 in vivo, we performed chimeric enhanced cross-linking immunoprecipitation-sequencing (eCLIP-seq) to identify

miR-486 muscle transcripts that may influence muscle function in complementation with our bulk *mir-486* KO RNA-seq analysis. Whole TA muscles from WT mice were biopsied and Argonaute-2 (Ago2)-miRNA complexes were immunoprecipitated with miR-486 bound to target mRNA transcripts (Van Nostrand et al, 2016). miR-486–specific sequence adapters were then ligated to the miRNA–mRNA hybrid chimera molecules followed by de–cross-linking, and the miR-486–bound transcripts were amplified into a cDNA library (Fig 5A). The library was sequenced using *mir-486* KO TA muscles as a control, revealing *miRNA-486*–bound targeted transcripts. The chimeric eCLIP-seq peak tracks demonstrate the chromosomal location of the 18 differentially expressed transcripts throughout the mouse genome (Fig 5B). The sequencing peaks of one of the identified transcripts *Mt2* are highlighted using the Integrated Genome Viewer sequencing visualization tool (Fig 5B). Chimeric eCLIP-seq revealed miR-486 binding to many 3'UTRs but also interestingly to a large numbers of coding sequences (CDS) in muscle genes (Fig 5C and D). The top 10 peaks that represent miR-486:mRNA bound complexes identified through chimeric eCLIP-seq analysis represent binding sites are listed in Fig 5E, and the full list of identified peaks is shown in Table S1. We performed functional

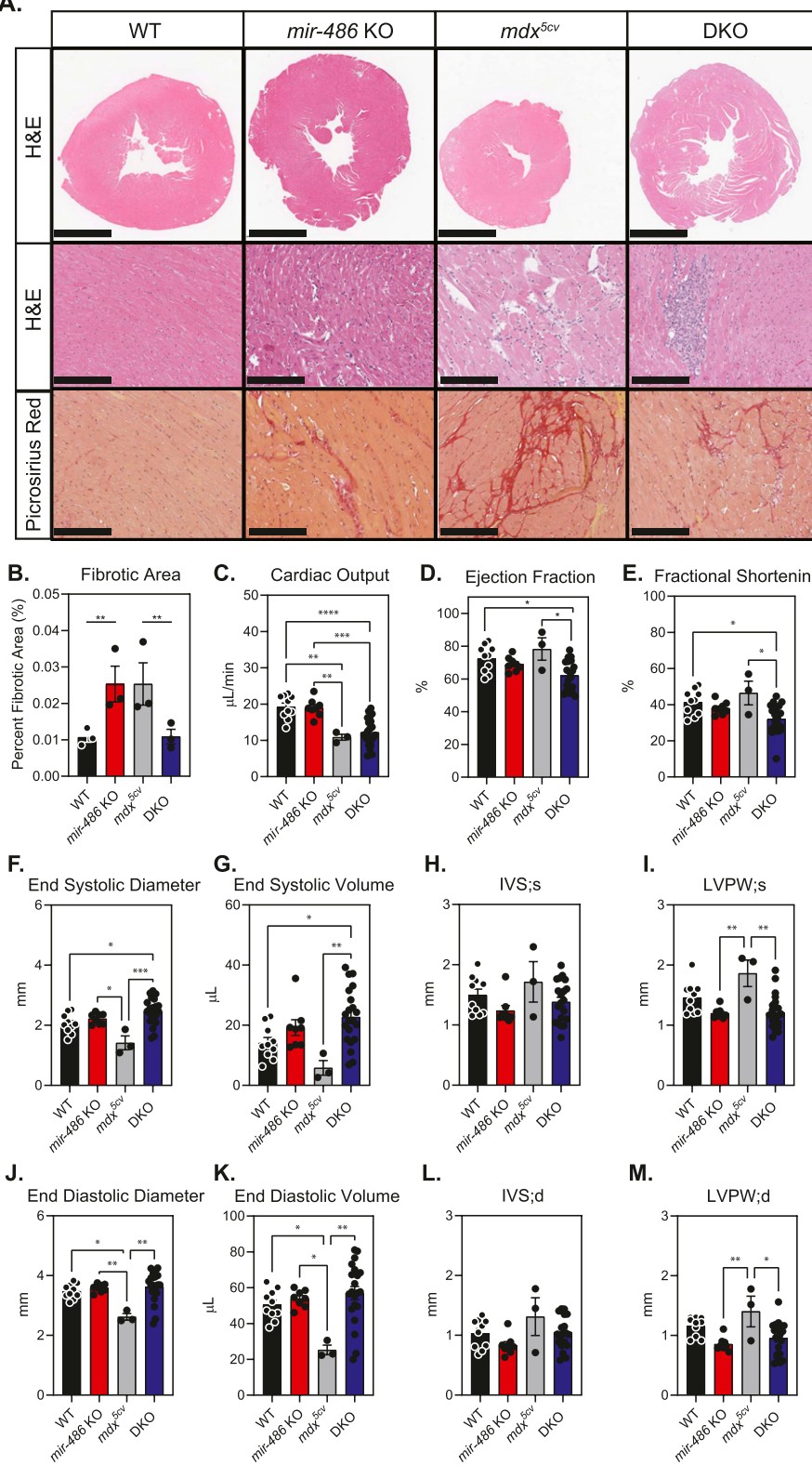

**Figure 3. *mir-486* KO mice develop functional and histological cardiac dysfunction.**
**(A)** Top row: representative hematoxylin and eosin-stained transverse sections of isolated left ventricle (LV). Scale bar = 2 mm. Middle row: representative hematoxylin and eosin-stained left ventricular myocardial sections. Scale bar = 300 *μM*. Bottom row: representative picrosirius red–stained images of left ventricular myocardial sections. Scale bar = 200 *μM*. Dark red areas indicate fibrotic tissue in picorosirus-stained sections. Sections cut at a thickness between 50 and 200 *μm*. **(B)** Fibrotic area calculated from picrosirius red–stained sections using ImageJ (five images from three animals per genotype). **P ≤ 0.01. N = 3 mice per genotype cohort. **(C)** Cardiac output (*μL*/min). **(D)** Percent ejection fraction. **(E)** Percent fractional shortening. **(F)** End systolic diameter (mm). **(G)** End systolic volume (*μL*). **(H)** IVS; s = intraventricular septum thickness during systole (mm). **(I)** LVPW; s = left ventricular posterior wall thickness during systole (mm). **(J)** End diastolic diameter (mm). **(K)** End diastolic volume. **(L)** IVS; d = intraventricular septum thickness during diastole (mm). **(M)** LVPW; d = left ventricular posterior wall thickness during diastole (mm). Approximately 8–20 mice per genotype were analyzed. Data are represented as mean ± SEM. *P ≤ 0.05, **P ≤ 0.01. Cardiac function parameters were obtained using VisualSonics small animal echocardiogram equipment and analyzed using VevoLab software.

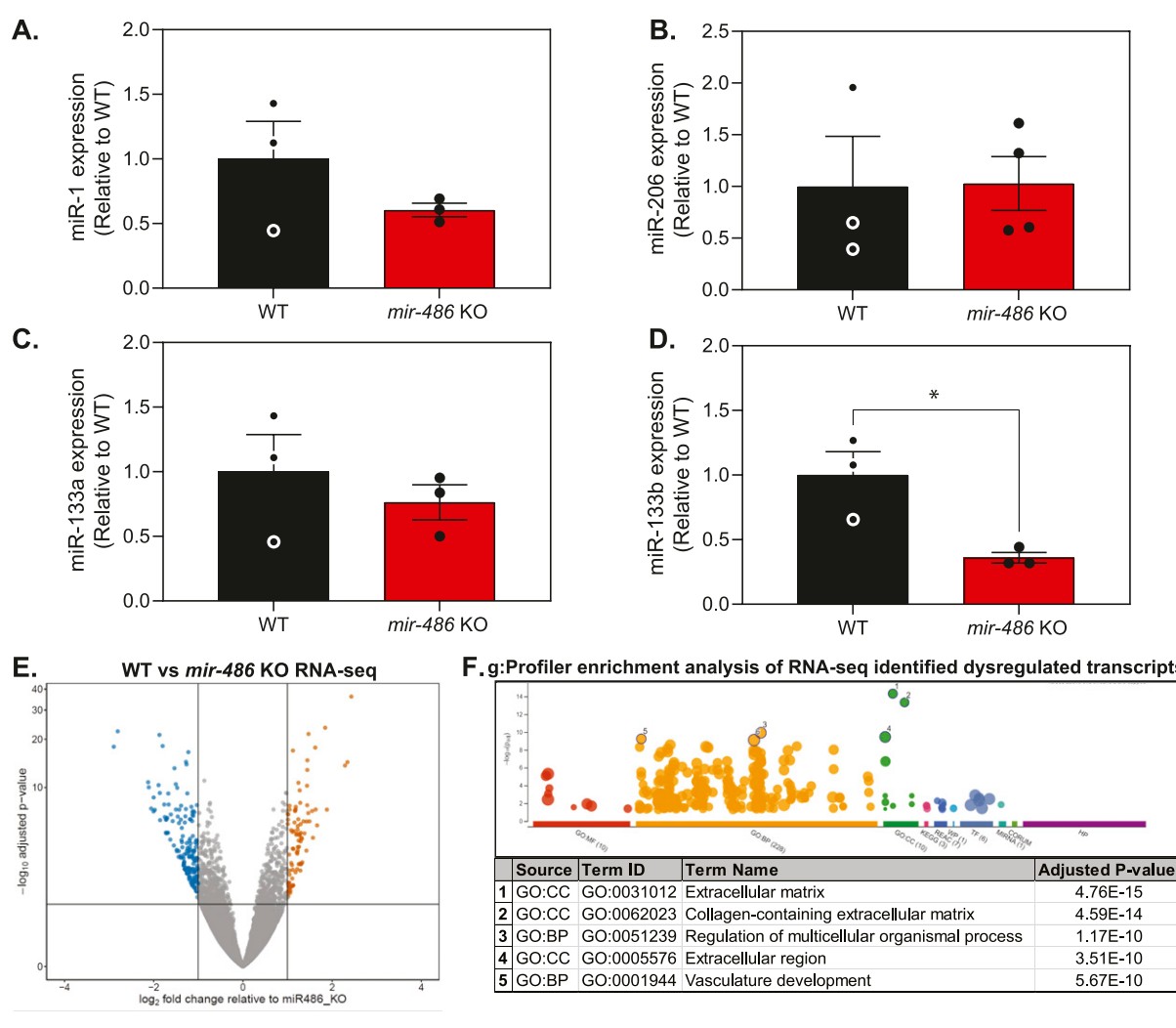

**Figure 4. RNA-seq reveals extracellular matrix pathways in *miR-486* KO muscle compared with WT.**
**(A, B, C, D)** Quantitative PCR reveals differential expression of myomiRs in 6-mo-old *mir-486* KO tibialis anterior (TA) muscle compared with WT. n = 3 mice per genotype, replicated in triplicate. **(E)** Volcano plot demonstrating the fold change and significance of differential gene expression in 6-mo-old *mir-486* KO TA muscle compared with WT controls. N = 5 mice/cohort were used for comparative analysis. N = 4 mice per genotype. **(F)** g:Profiler enrichment analysis of 85 transcripts increased in expression, and

enrichment analysis using the g:Profiler tool set to identify key gene ontology pathways of miR-486 muscle chimeric eCLIP-seq targets (Raudvere et al, 2019). The g:Profiler analysis revealed significant enrichment of miR-486 binding to contractile fiber, myofibril, sarcomere, and other muscle-associated targets as shown by the g: GOSt multi-query Manhattan plot (Fig 5F). We next performed quantitative PCR expression analysis on the 18 miR-486 targets (Table S1) identified through eCLIP-seq analysis and compared those to the transcriptome from our *mdx*[5cv] dystrophic mouse TA muscles. The goal was to correlate overlapping DMD biomarkers as many of the miR-486 eCLIP-seq targets had been shown to be dysregulated in DMD patients and involved in key ECM remodeling and metabolic pathways (Grounds et al, 2020). Unbiased analysis of the 18 miR-486 in vivo target transcripts identified via eCLIP-seq showed that many of them are early stage DMD biomarkers (Hathout et al, 2014, 2015; Guiraud et al, 2017) (Figs 5E and G and 3 and Table S1). Interestingly, although nearly all of the mRNA transcripts were induced in the *mir-486* KO muscles, several transcripts including those with the predicted miR-486 seed sites (e.g., *Mt2*) showed variable levels of induction in the absence of dystrophin (Fig 5G). The miR-486 targets that are decreased in expression in DMD muscle may be because of the more severe muscle pathologies and overall poor muscle quality observed in the *mdx*[5cv] mice as compared with the *mir-486* KO mice at 6 mo of age as the younger *mir-486* KO mice showed no significant muscle pathological remodeling compared with wild-type and *mdx*[5cv] age-matched cohorts (data not shown). We then performed a combined analysis of the *mir-486* KO versus WT TA muscle bulk RNA-seq, miR-486 chimeric eCLIP-seq, and *mdx*[5cv] versus WT TA muscle datasets to identify combined miR-486 target transcripts that may yield additional DMD disease biomarkers (Table S2). We noted that many of the miR-486 target transcripts identified from both bulk *mir-486* KO RNA-seq and miR-486 chimeric eCLIP-seq datasets shared common overlapping dysregulated transcripts with the *mdx*[5cv] bulk RNA-seq dataset, which suggested that reduced miR-486 expression may drive some of the pathological remodeling symptoms observed in our DMD mouse model (Table S3). To further define the mechanistic link between miR-486 transcriptomic regulation and the levels of dystrophin, we performed adeno-associated virus (AAV)–mediated shRNAi knockdown of the large dystrophin (Dp427) in primary differentiated mouse myotubes. The AAV-mediated Dp427 transcript shRNAi knockdown was shown to have a corresponding knockdown in miR-486 expression levels (Fig S4). Previous transcriptomic changes in AAV-mediated Dp427 shRNAi knockdown demonstrated significant alterations in muscle differentiation and structural transcripts as a consequence of mouse Dp427 protein loss (Seno et al, 2008, 2010). We then sought to further compare the overlap of dysregulated muscle transcripts in our miR-486 bulk KO and chimeric eCLIP-seq experiments with that of our *mdx*[5cv] RNA-seq to identify commonalities among the platforms with respect to dystrophic remodeling target transcripts. The miR-486 chimeric eCLIP-seq also identified several miR-486–associated

target transcripts that also were somewhat dysregulated in the *mdx*[5cv] versus WT TA muscle dataset but to a lesser extent (Table S3). This phenomenon may be because of either the lower levels of miR-486 in adult TA muscles (as opposed to higher levels in embryonic muscle) or because of the progressive nature of dystrophin-deficiency from which skeletal muscle is replaced with fibrotic or necrotic tissue. Nevertheless, many of the miR-486 targets identified using both bulk RNA-seq and chimeric eCLIP-seq all have been shown to play important roles in different parts of the dystrophic disease progression, which was consistent with our hypothesis, that miR-486 is an influencer of dystrophic disease progression. This approach of combining bulk RNA-seq with chimeric eCLIP-seq to identify targets of a key miRNA driving dystrophic disease pathology is useful towards identifying key transcriptomic pathways driving dystrophic muscle pathological remodeling.

## Discussion

The dynamic nature of miR-486 expression and its role in both pathological muscle remodeling and metabolic regulation suggests its requirement in muscle during times of stress and disease. Our findings demonstrate that miR-486 is a key DMD biomarker whose down-regulation drives pathogenic remodeling of dystrophic muscle by targeting key muscle structural, metabolic, and extracellular matrix remodeling factors. Indeed, miR-486 is required for normal skeletal and cardiac muscle growth and function through modulation of these factors as well. Loss of *mir-486* has profound consequences on mouse skeletal and cardiac muscle architecture, fibrotic depositions, centralized myonuclei, and overall muscle function. In cardiac muscle, *mir-486* ablation resulted in significant pathological remodeling of the heart and overall decreased cardiac output. There is growing evidence that restoration of dystrophin via exon-skipping compounds can restore the transcriptome signature of dystrophic muscle, including miRNAs, to that of normal or Becker-like muscle (Coenen-Stass et al, 2015). In Duchenne muscular dystrophy (DMD), the lack of a functional dystrophin protein causes skeletal muscle weakness, dilated cardiomyopathy, respiratory failure, and premature death (Duan et al, 2021). Recent progress in the development of *DYSTROPHIN* gene restoration approaches has been shown to benefit a subset of DMD patients amenable to the skipping of *DYSTROPHIN* exon 51 which restores the reading frame (Charleston et al, 2018). Other *DYSTROPHIN* gene–independent strategies involve the overexpression of a truncated micro-dystrophin (μDYS) construct via adeno-associated viral (AAV) vectors that are under clinical development (Crudele & Chamberlain, 2019). The culmination of these strategies centers on improvement of dystrophic pathologies in skeletal and heart muscle but does not address the systemic effects that occur because of dystrophin protein loss. Our study also demonstrates that miR-486 expression is tightly regulated by myogenic factors, and its decrease in expression can be correlated with validated dystrophic

---

159 transcripts decreased in expression based on ≥ 2.0 log$_2$ fold change of WT versus *mir-486* KO RNA-seq analysis. The top 5 pathway hits are listed below the graph. **(G)** Table of top 10 transcripts increased in expression in 6-mo-old *mir-486* KO mouse TA muscle compared with WT as identified by RNA-seq. **(H)** Table of top 10 transcripts decreased in expression in 6-mo-old *mir-486* KO mouse TA muscle compared with WT as identified by RNA-seq.

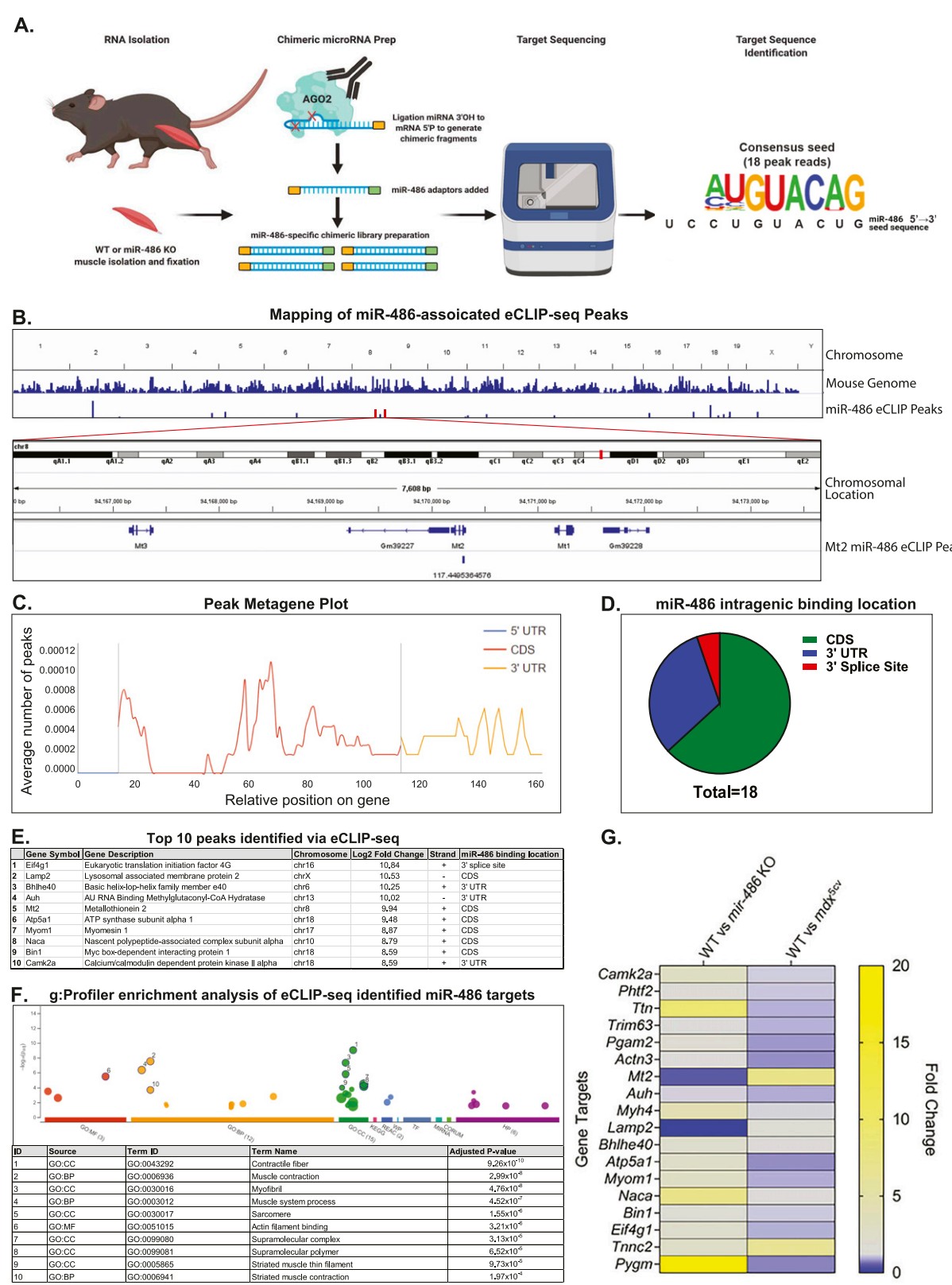

**Figure 5. Targeted chimeric miR-486 eCLIP-seq in WT and *mir-486* KO muscles.**
**(A)** Schematic demonstrating the workflow for the chimeric eCLIP sequencing platform to identify miR-486 in vivo skeletal muscle regulated transcripts. Tibialis anterior muscles were harvested from 6-mo-old WT and *mir-486 KO* male mice, and total RNA isolation was completed. The Ago2-miR-486 complex bound to target RNA transcripts was isolated, and then sequencing was performed to map the miR-486–associated peak tracks. The sequence logo (AUGUACAG) represents the consensus

biomarkers. Furthermore, the reduction of miR-486 expression is tightly correlated with dystrophic disease progression, thus making it a strong candidate DMD biomarker for monitoring dystrophic symptoms and muscle pathological remodeling.

Our combinatorial RNA sequencing and chimeric eCLIP-seq of skeletal muscles from *mir-486* KO mice identified a network of miR-486 muscle targets including coding and noncoding regions of muscle targets that play key roles in dystrophin-deficient muscle pathology. miRNA profiling studies demonstrated that miR-486 levels increased in the right ventricles of hypoplastic left heart syndrome (HLHS) patients, and miR-486 is elevated in response to cyclic stretch (Lange et al, 2019). Other studies demonstrate that miR-486 levels are modulated during development, stress, and disease states via modulation of key cellular growth pathways through direct regulation of FOXO, SMAD, PI3K, and AKT (Alexander et al, 2011, 2014; Margolis et al, 2018; Lange et al, 2019). These growth pathways are also dysregulated in dystrophic skeletal muscles and can be targeted therapeutically via miR-486 transgenic over-expression in skeletal muscle to restore proper signaling pathways (Alexander et al, 2011, 2014).

Additional questions remain as to which specific myogenic cell populations and during which stages of dystrophic development miR-486 dysregulation occurs. Several studies have identified miR-486 enrichment in quiescent muscle satellite cells, along with other muscle-enriched myomiRs before activation (Mizbani et al, 2016). As activation of muscle satellite cells occurs along with myogenic differentiation, miR-486 expression increases which may represent the switch from MyoD-driven miR-486 expression to SRF-driven expression that we show at the *mir-486/ANK1-5* locus. Although miR-486 expression was shown to regulate *Pax7* in quiescent mouse muscle satellite cells, there are likely additional miR-486 targets in MSCs and other nonmuscle cell types. Furthermore, many of the miR-486 mRNA targets we identified are also expressed in nonmuscle cell types. This suggests that additional levels of posttranscriptional regulation may yield novel tissue-specific miR-486 target transcripts that have yet to be identified.

miRNA-486 overexpression may be a valid means of restoration of key dysregulated signaling pathways in dystrophic muscle that affect growth and muscle function. There is mounting evidence that the collagens and the ECM remodeling that occurs in DMD serves as a biomarker for disease progression and a therapeutic target (Regensburger et al, 2019; Strandberg et al, 2020; Signorelli et al, 2021). Additional evidence demonstrating the restoration of miRNAs in exon-skipped *mdx* mice suggests that miR-486 expression profiling is highly useful in evaluating dystrophic replacement and restoration strategies in muscle (Roberts et al, 2015, 2016; Coenen-Stass et al, 2018; van Westering et al, 2020). Given our previous work demonstrating that miR-486 transgenic overexpression in a dystrophic mouse model can ameliorate dystrophic pathologies, strategies to either induce miR-486 expression or direct miR-486 overexpression should be explored for therapeutic disease modulation.

# Materials and Methods

## Human subjects

Patient muscle biopsies from normal, DMD, and Becker muscular dystrophy patients were obtained from fully consented patients as previously described (Reid et al, 2020). All human subject protocols were approved by the University of Alabama at Birmingham (UAB) Institutional Review Board under the protocol 300002164.

## Mice

The *mir-486* KO mice were generated as previously described (Jee et al, 2018). Mice were initially maintained on the *FVB/NJ* (#001800; Jackson Labs) strain and were later backcrossed onto the *C57BL/6J* (#000664; Jackson Labs) strain for more than six generations to ensure isogenicity. All analyses were performed on the same strain background. The *mdx$^{5cv}$* (#002379; Jackson Labs) were also kept on the *C57BL/6J* strain, and homozygote mutant females were mated to the *mir-486* KO males to generate *mir-486+/−:mdx$^{5cv}$* mice and later *mir-486* KO:*mdx$^{5cv}$* (DKO) mice. Unless otherwise stated, only 6-mo-old male mice were used in all experimental conditions. All mouse strains were maintained under standard housing and feeding conditions with the University of Alabama at Birmingham Animal Resources Facility under pathogen-free conditions, and all pro-tocols were approved under the animal protocol number 21393.

## Chromatin immunoprecipitation (ChIP)

Chromatin immunoprecipitation was performed on normal human and DMD myoblasts and myotubes in a protocol previously de-scribed (Alexander et al, 2013). Genomic DNA was isolated from myoblasts (70% confluency in growth medium) and myotubes (4 d differentiation in low serum differentiation medium), and ChIP was performed using the Simple ChIP Enzymatic Chromatin IP Kit (Cat. no. 9003; Cell Signaling Technology) following the manufacturer's protocols. For the MyoD ChIP assays, a rabbit polyclonal (MyoD clone M-318; Cat. no. sc-760; Santa Cruz Biotechnology) was used to immunoprecipitate MyoD bound to the mouse *mir-486* locus. For SRF ChIP assays, a rabbit polyclonal (SRF clone G-20; Cat. no. sc-335; Santa Cruz Biotechnology) was used to immunoprecipitate SRF bound to the mouse *miR-486* locus. Myoblasts were seeded at 4 ×

---

sequence for the top 18 miR-486–associated peak reads based on the miRNA:mRNA target sequencing alignments. The miR-486 seed sequence is shown in the 5′–3′ direction. **(B)** Chromosomal location of a single top hit peak transcript, *Mt2*, as identified by eCLIP-seq as a direct target of miR-486. Peaks generated using Integrative Genome Viewer. **(C)** Metagene plot demonstrating overall miR-486 binding location by a relative position on the target gene. **(D)** Pie chart demonstrating the proportion of miR-486 gene targets and the respective intragenic binding location of miR-486. **(E)** Table outlining the top 10 transcripts that were identified as direct targets of miR-486 via CLIP-seq. **(F)** g:Profiler enrichment analysis graph demonstrates the most significant cellular pathways associated with the 18 direct miR-486 targets identified via chimeric eCLIP-seq. The pathway ID number in the table correlates with the numbered dots in the accompanying graph above. **(G)** Heat map expression of quantitative PCR of 18 miR-486 eCLIP-seq targets in *mir-486* KO and *mdx$^{5cv}$* tibialis anterior muscles expression levels compared with WT controls in separate cohort analyses. Data points are individual biological replicates, N = 4/cohort, and logarithmic fold change normalized to both wild type and *β*-actin is shown. **P ≤ 0.01 N = 3 replicates per cohort. Transcript levels are normalized to *β*-actin, and *mir-486* KO levels are shown as relative to WT.

$10^5$ cells/15 cm dish in 10 dishes per cohort. The myoblast fraction consisted of normal human myoblasts grown to ~70% confluency. For the myotube fraction (five 15 cm dishes), after reaching 90% confluency, myoblasts were differentiated into myotubes by serum withdrawal (2% FBS) for 4 d. Cells were harvested after three washes in ice-cold 1× DPBS, were cross-linked with 1% formaldehyde (Cat. no. F8775; Sigma-Aldrich) for 1 h at room temperature, and then chromatin was sheared using by incubating the DNA pellets with micrococcal nuclease for 20 min at 37°C. Ten percent of chromatin was removed for input controls, whereas ~200 mg of total chromatin was used to purify DNA fragments after immunoprecipitation with 5 mg of either MyoD or SRF antibodies. The following morning, the samples were incubated with 30 ml of ChIP Grade Protein G magnetic beads (Cat. no. 9006S; Cell Signaling Technology) for 2 h at 4°C under gentle rotation. After several stringent washes, the chromatin was eluted off the columns in 1 mM Tris–HCL pH 8.0 (Cat. no. 15568025; Thermo Fisher Scientific) and purified on DNA-binding columns (Cat. no. D4029; Zymo Research) before a final elution in sterile water. MyoD and SRF ChIP-seq annotation of ANK1-5 was completed using the publicly available UCSC Genome Browser track data hubs annotation resource available at http://genome.ucsc.edu. For the MyoD TF binding site at the human *ANK1-5* locus, the following primers were used Fwd: 5'-GAGCTCCAA-GACTGAGGACTGGAC-3' and Rev: 5'-CAGGGAGGATGGAGATCAGAGCC-3'. The nonspecific MyoD site negative control primers next to the human *ANK1-5* locus used were Fwd: 5'-CTGCACGTCAGCCTCCCAAAG-3' and Rev: 5'-ACTGGGATCCTCCAGGGGCC-3'. The MyoD TF positive-control primers used were Fwd: 5'-CAGTGAACAATGGTGCTTGG-3' and Rev: 5'-TTCCACATTCACGCAGAGAG-3'. For the SRF TF-binding site at the human *ANK1-5* locus, the following primers were used Fwd: 5'-ACAGTAGGTGAGTTGCAGGGTTAG-3' and Rev: 5'-GGGCTCAGG-GACAGTCAAGTGAGC-3'. The nonspecific SRF site negative-control primers next to the human *ANK1-5* locus used were Fwd: 5'-GAAACACGGAGCAGCCTGGC-3' and Rev: 5'-ATGAGGATCGACTGTA-CATGC-3'. The SRF TF positive-control primers used were Fwd: 5'-TGGTTGGATAACAGAGGCAGA-3' and Rev: 5'-GCTTCTGTTGTGG-CGTCTTT-3'.

## EdU myoblast proliferation assay

To assess the proliferation of myoblasts in *mir-486* KO mice versus WT cells were plated in four-well chamber slides (Nunc Lab Tek II; Thermo Fisher Scientific) previously coated with 0.01% gelatin at seeded at 5,000 cells/well for 24 h under growth medium conditions at 37°C. After overnight incubation, each chamber was stained with EdU-Alexa Fluor 488 probes using the Click-iT EdU Cell Proliferation Kit for Imaging (Thermo Fisher Scientific, Cat. no. C10337) according to manufacturer's instructions for 3 h and then imaged on a DS-Qi2 Nikon Microscope (Nikon Instruments Inc.).

## Western blots

Protein lysates from cell and tissues were lysed and homogenized (tissues) in Mammalian Protein Extraction Reagent (M-PER) lysis buffer (Cat. no. 78501; Thermo Fisher Scientific) supplemented with cOmplete protease inhibitor tablets (Cat. no. 1183617000; MilliporeSigma). Approximately 50 $\mu$g of lysate were electrophoretically resolved on 4–20% Novex Tris-glycine gradient gels (Cat. no. XV04200PK20; Thermo Fisher Scientific). Proteins were then transferred to 0.2-$\mu$m PVDF membranes (Cat. no. 88520; Thermo Fisher Scientific) and incubated overnight at 4°C with gentle rocking in primary antisera diluted 1:1,000 in 5% bovine serum albumin (Cat. no. A30075; RPI Corp)/1xTBS-Tween (Cat. no. IBB-581X; Boston BioProducts). Membranes were washed three times in 1xTBS-Tween for 5 min each and then incubated in secondary antisera (1:2,000 dilution) for 1 h at room temperature. Membranes were then washed four times in 1× TBS-Tween for 15 min each before the addition of the Novex Chemiluminescent Substrate Reagent Kit (Cat. no. WP20005; Thermo Fisher Scientific). Membranes were exposed onto PerfectFilm audioradiography film (Cat. no. B581; GenHunter) and developed on a Typhoon Variable Mode Imager (Amersham Pharmacia). Western blot densitometry was performed using open-source ImageJ software.

## Immunofluorescence of muscle sections

For immunofluorescent staining, frozen sections were fixed in 100% acetone for 5 min at room temperature, then air dried for 20 min. The slides were then incubated was in 1x PBS-Tween (Cat. no. IBB-171R; Boston BioProducts). To reduce nonspecific binding, the slides were then incubated with blocking reagent from the Mouse-on-Mouse kit (Cat. no. BMK-2202; Vector Laboratories). Slides were incubated for 1 h at room temperature in the primary antibody. Slides were imaged with a Zeiss LSM 700 Laser Scanning Confocal at ×10 objective, and Zen software (Carl Zeiss Microscopy, LLC) was used for image capture. ImageJ software was used to generate montages of muscle sections for fiber counting. Adobe Photoshop version 2020 (Adobe Systems Inc.) was used to adjust resolution and contrast for representative images.

## Muscle histochemical staining

Mouse skeletal muscle tissues were cryo-frozen by covering the tissue Optimum Cutting Temperature solution (Thermo Fisher Scientific; Cat. no. 23-730-571) and completely submerging the tissues in a liquid nitrogen–chilled isopentane (Cat. no. AC397221000; Thermo Fisher Scientific) bath as unfixed tissues. Blocks were later cut on a cryostat and 7–10-$\mu$m thick sections were placed on Superfrost Plus slides (Cat. no. FT4981GLPLUS; Thermo Fisher Scientific). Hematoxylin and eosin (H&E) staining was performed as previously described (Alexander et al, 2011). Mouse skeletal muscle tissues were cryo-frozen and sectioned as described above. Masson's Trichrome staining was performed on frozen sections using the Masson's Trichrome Stain Kit (Cat. no. 25088-100; Polysciences) following the manufacturer's protocol. Mouse hearts were perfusion fixed in 10% neutral buffered formalin (Cat. no. HT501128; MilliporeSigma) overnight at 4°C and held in 70% ethanol for 24–48 h before being processed as previously described (Reid et al, 2020).

## RNA sequencing and data analyses

Adult 6-mo-old male TA muscles from WT and *mir-486* KO mice (n = 4 mice per cohort) were snap frozen in liquid nitrogen. Muscle samples were mechanically homogenized, and total RNA was

extracted using a miRVana Isolation Kit (Cat. no. AM1560; Thermo Fisher Scientific) following the manufacturer's protocol. The total RNA was amplified using the Sure Select Stranded RNA-Seq kit (Agilent Technologies) using standard protocols. A ribominus kit (Cat. no. K155002; Thermo Fisher Scientific) was used to deplete large ribosomal RNAs. All biological replicates contained a minimum of 35.7 million reads with an average number of 39.6 million reads across the replicates. The FASTQ files were uploaded to the UAB high-performance computer cluster for bioinformatics analysis with the following custom pipeline built in the Snakemake workflow system (v5.2.2) (Koster & Rahmann, 2012): first, quality and control of the reads were assessed using FastQC, and trimming of the Illumina adapters and bases with quality scores of less than 20 were performed with Trim Galore! (v0.4.5). After trimming, the transcripts were quasi-mapped and quantified with Salmon (Patro et al, 2017) (v0.12.0, with "--gencode" flag for index generation and "-l ISR," "--gcBias," and "--validateMappings" flags for quasi-mapping) to the mm10 mouse transcriptome from Gencode release 21. The average quasi-mapping rate was 70.4%, and the logs of reports were summarized and visualized using MultiQC (Ewels et al, 2016) (v1.6). The quantification results were imported into a local RStudio session (R version 3.5.3), and the package "tximport" (Soneson et al, 2015) (v1.10.0) was used for gene-level summarization. Differential expression analysis was conducted with DESeq2 (Love et al, 2014) package (v1.22.1). After count normalization, principal component analysis was performed, and genes were defined as differentially expressed genes if they passed a statistical cutoff containing an adjusted $P$-value < 0.05 (Benjamini–Hochberg false discovery rate method) and if they contained an absolute $\log_2$ fold change $\geq$ 1. Functional annotation enrichment analysis was performed in the National Institutes of Health (NIH) Database for Annotation, Visualization and Integrated Discovery (DAVID, v6.8) by separately submitting up-regulated and down-regulated differentially expressed genes. A $P$-value < 0.05 cutoff was applied to identify Gene Ontology terms. The FASTQ files of the current study have been uploaded to National Center for Biotechnology Information (NCBI)'s Gene Expression Omnibus under accession number GSE155787.

## Chimeric eCLIP Sequencing

For the chimeric CLIP-sequencing, we used adult 6-mo-old male TA muscles from WT and *mir-486* KO mice (n = 3 muscles for each cohort) that were snap-frozen in liquid nitrogen, and chimeric eCLIP sequencing was performed with assistance by Eclipse Bioinnovations (Eclipse Bioinnovations). The FASTQ files of the current study have been uploaded to NCBI's Gene Expression Omnibus under accession number GSE173821. A detailed description of the chimeric CLIP-seq procedures including Ago2 immunoprecipitation, library amplification, and other quality control analyses can be found in the Supplementary Figures and the Materials and Methods section.

## Real-time quantitative PCR (qRT-PCR)

Total RNA was extracted from muscle tissue using the miRVana (Cat. no. AM1560; Thermo Fisher Scientific) kit following the manufacturer's protocol. One microgram of total RNA was reverse-transcribed using a TaqMan Reverse Transcription Kit following the manufacturer's protocol (Cat. no. N8080234; Applied Biosystems). For miRNA fractions, 50 ng of total small RNA was used for all reverse-transcription reactions. TaqMan assay probes were purchased from Applied Biosystems corresponding to the individual genes. Quantitative PCR (qPCR) TaqMan reactions were performed using TaqMan Universal PCR Master Mix (Applied Biosystems; Cat. no. 4304437). Samples were run on the Fluidigm Biomark HD system (Fluidigm Corp.) in 96.96 Dynamic Array plates. Relative expression values were calculated using the $2^{-\Delta\Delta Ct}$ method (Livak & Schmittgen, 2001).

## Echocardiograms

Echocardiograms were performed on mice individually by the UAB Mouse Cardiovascular Core using the VisualSonics Vevo 3100 Imaging and Ultrasound system (FUJIFILM VisualSonics). Ultrasound echocardiograms were analyzed using VisualSonics VevoLab 3.2.0 software. Data were analyzed and graphed using GraphPad Prism version 9 (GraphPad Software).

## Statistical analyses

Unless otherwise stated, all statistical comparisons of multiple-variables (e.g., time and genetic cohort) were performed using two-way ANOVA, whereas single variable (e.g., genetic cohort) were performed using one-way ANOVA. All ANOVA tests were corrected with a Tukey HSD test. Values of significance are stated within the figure legends. Numbers of replicates used are directly stated in the figure legend or the Materials and Methods section. Real-time quantitative-PCR validation statistical analyses used two tailed unpaired $t$ tests to determine significance, and results are presented as mean ± SEM. Differences were considered statistically significant at $P \leq 0.05$. All statistical tests were made with GraphPad Prism version 9.

## Dystrophin shRNAi knockdown

Mouse *Dystrophin* (Dp427 transcript) was targeted in primary WT mouse myotubes 4 d in differentiated medium (D4) using an adeno-associated viral (AAV) shRNAi vector (pAAV-MCS8-IZsGreen; gift from Jeng-Shin Lee Harvard Viral Core) from with (5′-GGCCUUA-CAGGGCAAAAAC-3′ sense; 5′-GUUUUUGCCCUGUAAGGCC-3′ anti-sense) and cloned into the vector using the EcoR1/Xba1 restriction sites. The shRNAi hairpin design was previously shown to target the mouse Dp427 transcript (exons 3–4) (Seno et al, 2008). AAV shRNAi-Dp427, AAV-shLuc (targeting luciferase; 5′-TTTGTGAAGCAGATGAAA-TAG-3′ sense; 5′-CTATTTCATCTGCTTCACAAA-3′ antisense), or mock (1xDBS) was transduced at MOIs of 1, 100, and 200 ($10^5$ vector genomes/$\mu$l for an MOI = 1) into primary mouse myotubes, and muscle lysates were harvested 72 h post infection. Viral titers were concentrated at ~$10^9$ vector genomes/$\mu$l. Myotubes were lysed using M-PER lysis buffer (Cat. no. 78501; Thermo Fisher Scientific) supplemented with protease inhibitor tablets and resolved on 4–20% Novex Tris-glycine gradient gels as described in the main Materials and Methods section. Immunoblotting antisera against dystrophin

(Cat. no. ab15277; Abcam) and $\beta$-tubulin (Cat. no. 2128S; Cell Signaling technology) demonstrates specific knockdown of the mouse *Dystrophin* Dp427 transcript. TaqMan probes (Thermo Fisher Scientific) were used against mouse Dp427 (exons 16–17), miR-486, RNU6-2, and Ank1-5 to detect changes in transcript levels.

### Activity tracking and treadmill running

Activity levels were tracked using the Noldus Ethovision XT12 and 30 × 30 cm open-field chambers in a 2 × 2 set-up. Mice were given 5 min to assimilate to the open-field chambers, and pre-exercise activity was recorded via live video tracking, resulting in total distance traveled, measured in cm. Mice were then immediately taken to the treadmill for testing. Mice were run on a 10-degree decline on an Exer-3/6 treadmill (Columbus Instruments) treadmill (UAB Neurobehavioral Core) for 10 min per run. Mice were run in a speed-progressing manner: 5 m/min for 1 min, 6 m/min for 1 min, 7 m/min for 1 min, 8 m/min for 1 min, 9 m/min for 1 min, then finally 10 m/min for 5 min. Immediately after treadmill, mice were again placed in the open-field chambers, and postexercise activity was recorded.

### Quantitative magnetic resonance imaging

To evaluate body composition (fat and lean tissue mass) in vivo, 6-mo male WT and *mir-486* KO mice (10 mice/genotype) were measured using the EchoMRI 3-in-1 composition analyzer (software version 2016; Echo Medical). Individual fat and lean mass measurements were standardized to total body weight, and *mir-486* KO values were statistically compared with WT values using a one-way ANOVA with a Tukey's post hoc correction.

### Physiological force measurements

The experimental mice were anesthetized with sodium pentobarbital (intraperitoneal injection of 80 mg/kg) for dissection of the extensor digitorum longus (EDL) (euthanasia via pneumothorax). The EDL was maintained ex vivo in a bicarbonate buffer (35°C, equilibrated with 95% $O_2$, 5% $CO_2$) and assayed for tetanic force per muscle physiological CSA (specific force) as previously described (Alexander et al, 2014). All statistical data were analyzed by one-way ANOVA followed by evaluation of significant treatment effects with Tukey's HSD post hoc test. The type I error rate was set at $P$-value < 0.05.

### Chimeric eCLIP sequencing library preparation

Whole TA muscles were isolated from male 6-mo-old wild-type and *mir-486* KO mice (n = 3 total samples per genotype; performed in duplicate) and snap-frozen in liquid nitrogen. The samples were processed for chimeric eCLIP sequencing by Eclipse Bioinnovations Inc. Approximately 80 mg of mouse muscles were homogenized by cryogenic pulverization. Samples were then were UV (254 nm) cross-linked twice at 400 mJ/cm2 using Stratalinker 2400 (Stratagene), on a bed of ice. After cross-linking, the samples were sonicated (QSonica Q800R2; QSonica LLC) to shear the genomic DNA into smaller fragments. Ago2 immunoprecipitation using an Ago2 antibody (Eclipse Bioinnovations Inc.) was pre-coupled to anti-

mouse Dynabeads (M-280 Sheep Anti-Mouse IgG Dynabeads, 11201D; Thermo Fisher Scientific), added to the homogenized lysate, and incubated overnight at 4°C with gentile rocking. After immunoprecipitation, 2% of the sample was taken as the paired input sample. For chimeric eCLIP experiments, a standardized eCLIP protocol (Van Nostrand et al, 2016) was modified to enable chimeric ligation of the miRNA and mRNA (Manakov et al, 2022 *Preprint*). miRNA-specific chimeric eCLIP was performed by amplifying cDNA using a mouse miR-486a primer and a sequencing adapter-specific primer, after amplification with indexed primers.

### Western blot of immunoprecipitation during AGO2 eCLIP in WT and mir-486 KO cells

A total of 10% of IP samples and 1% of input samples were run on NuPAGE 4–12% Bis-Tris protein gels, transferred to PVDF membrane, and probed with 1:1,000 of AGO2 WB antibody and 1:10,000 TrueBlot anti-Rabbit IgG (HRP) (Rockland Immunochemicals) and imaged with a C300 Imager for 1 min on normal settings using Azure Radiance ECL. Only the region from 85 to 160 kD (protein size to 75kD above) was isolated during chimeric eCLIP.

### Chimeric eCLIP sequencing analysis pipelines and software used

#### *Custom scripts*
count_aligned_from_sam.py
make_bigwig_files.py
normalize_bedGraph.py
negBedGraph.py
fix_scores.py
Peak_input_normalization_wrapper.py
compress_l2foldenrpeakfi_for_replicate_
overlapping_bedformat.pl
compress_l2foldenrpeakfi_for_replicate_overlapping_
bedformat_outputfull.pl
overlap_peakfi_with_bam_SE.pl
annotate_peaks_bedformat_
wproxdistal_lncRNA.pl
annotate_peaks_fullformat_wproxdistal_lncRNA.pl
make_informationcontent_from_peaks.pl
run_and_parse_IDR.pl
get_reproducing_peaks.pl
merge_idr_entropy.pl

#### *Other programs used*
Clipper: https://github.com/YeoLab/clipper
FastQC: v. 0.11.5
Cutadapt: v. 1.15
STAR: v. 2.6.0c
Samtools: v. 1.5
bedToBigBed: v. 332
bedGraphToBigWig v. 332
Bedtools: v. 2.26.0
R: v. 3.4.1

#### *Python and Python package versions*
Python v. 3.6.3::Anaconda v. 4.3.30 (64-bit)

Pysam v. 0.11.2.2
Htseq v. 0.9.1
Numpy v. 1.13.3
Pandas v. 0.21.0
Pybedtools v. 0.7.10
Scikit-learn v. 0.19.1
Scipy v. 1.0.0
Matplotlib v. 2.0.2
Gffutils v. 0.9
Seaborn v. 0.8.0
Statsmodels v. 0.8.0

**Perl packages used**
Statistics::Distributions v. 1.02
Statistics::Basic v. 1.6611
Statistics::R v. 0.34

**Packages used in Clipper conda environment**
Python v. 2.7.11::Anaconda v. 4.3.30 (64-bit)
Bedtools v. 2.26.0
Gffutils v. 0.8.2
HTSeq v. 0.6.1
Matplotlib v. 1.5.1
Numpy v. 1.11.1
Pandas v. 0.21.0
Pybedtools v. 0.7.2
Pysam v. 0.10.0
Samtools v. 1.3.1
Scikit-learn v. 0.17.1
Scipy v. 0.17.0

The entire processing pipeline was performed by using the program Snakemake to procedurally perform all processing steps in order. An alternate conda environment, which is specified within the Snakefile, is used to run the Clipper step because Clipper requires older software versions.

# Supplementary Information

# Acknowledgements

The authors wish to thank Alexander Shishkin, Kylie Shen, Sergei Manakov, Heather Foster, and Gene Yeo from Eclipse BioInnovations for their performance and assistance with the chimeric eCLIP sequencing experiments. We wish to thank Margaret Bell, Kaleen Lavin, Layne Peacock, and Gina Seay for assistance with the collection of normal human muscle control biopsies. We wish to thank Emanuela Gussoni and Glenn Rowe for a critical reading of the manuscript. RM Hightower is a funded by an National Institutes of Health (NIH) National Institute of Neurological Disorders and Stroke (NINDS) F99/K00 grant (F99NS118718). MA Lopez is funded by an NIH NINDS K08 grant NS120812 and a grant from the Kaul Pediatric Research Institute. The University of Alabama at Birmingham (UAB) Small Animal Phenotyping Core supported by the NIH Nutrition & Obesity Research Center (P30DK056336) and the Mouse Cardiovascular Core Vevo 3100 Mouse Ultrasound Facility for this project. Research reported in this publication was supported by Eunice Kennedy Shriver National Institute of Child Health and Human Development, NIH, HHS of the National Institutes of Health under award number R01HD095897 awarded to MS Alexander. MS Alexander is also a co-investigator on an NIH NIAMS award R21AR074006 and is also funded by a Muscular Dystrophy Association (MDA) grant (MDA418254). MM Bamman was supported by NIH NIAMS U01AR071133 and NICHD P2HD086851 grants. M Xie was supported by NIH NHLBI grants R01HL153501 and R03HL141620. Work in the EC Lai lab was supported by NIH grants R01GM083300 and R01HL135564 and by the MSK Core Grant P30CA008748.

## Author Contributions

A Samani: data curation, investigation, and writing—original draft, review, and editing.
RM Hightower: data curation, investigation, and writing—original draft.
AL Reid: data curation, investigation, and writing—original draft.
KG English: data curation, formal analysis, and investigation.
MA Lopez: resources, investigation, and writing—original draft, review, and editing.
JS Doyle: resources.
MJ Conklin: resources.
DA Schneider: supervision and writing—original draft, review, and editing.
MM Bamman: resources, supervision, and writing—original draft, review, and editing.
JJ Widrick: data curation, formal analysis, investigation, and writing—original draft, review, and editing.
DK Crossman: resources, data curation, formal analysis, and writing—original draft, review, and editing.
M Xie: resources, data curation, formal analysis, investigation, and writing—original draft, review, and editing.
D Jee: resources and writing—original draft, review, and editing.
EC Lai: resources, investigation, methodology, and writing—original draft, review, and editing.
MS Alexander: conceptualization, resources, data curation, formal analysis, supervision, funding acquisition, validation, investigation, methodology, project administration, and writing—original draft, review, and editing.

## Conflict of Interest Statement

The authors declare that they have no conflict of interest.

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
