## [Reviewer comments · Life Science Alliance]

Life Science Alliance

miR-486 is essential for muscle function and suppresses a dystrophic transcriptome

Adrienne Samani, Rylie Hightower, Andrea Reid, Katherine English, Michael Lopez, J. Doyle, Michael Conklin, David Schneider, Marcas Bamman, Jeffrey Widrick, David Crossman, Min Xie, David Jee, Eric Lai, and Matthew Alexander

DOI: <https://doi.org/10.26508/lsa.202101215>

Corresponding author(s): Matthew Alexander, University of Alabama at Birmingham

Review Timeline:

Submission Date:	2021-08-25
Editorial Decision:	2021-10-04
Revision Received:	2022-01-26
Editorial Decision:	2022-03-02
Revision Received:	2022-04-15
Editorial Decision:	2022-04-19
Revision Received:	2022-04-19
Accepted:	2022-04-20

Transaction Report:

October 4, 2021

Re: Life Science Alliance manuscript #LSA-2021-01215-T

Dr. Matthew Alexander
University of Alabama at Birmingham
Pediatrics
1918 University Blvd.
MCLM 922 #96
Birmingham, AL 35294

Dear Dr. Alexander,

Thank you for submitting your manuscript entitled "miR-486 is essential for muscle function and suppresses a dystrophic transcriptome" to Life Science Alliance. The manuscript was assessed by expert reviewers, whose comments are appended to this letter. We invite you to submit a revised manuscript addressing the Reviewer comments.

Thank you for this interesting contribution to Life Science Alliance. We are looking forward to receiving your revised manuscript.

Sincerely,

B. MANUSCRIPT ORGANIZATION AND FORMATTING:

Reviewer #1 (Comments to the Authors (Required)):

In this manuscript, Hightower and colleagues report on histological and molecular features of a newly generated miR486-KO mouse model. The findings are rather impactful for the overall field of muscle physiology and particularly relevant for the muscular dystrophy field. In that regard, the data from DMD and BMD patients' biopsies in Figure 1A are a striking and cogent rationale for the whole study. I would like to commend the Authors for bringing together histology, functional analyses, transcriptomics and eCLIP (miRNA target prediction based on physical miRNA binding and not computational prediction) to decipher the molecular and physiological relevance of this microRNA for muscle remodeling.

Here are some comments that I hope will help review and improve the present manuscript:

- it is interesting to note that the magnitude of miR-486 downregulation seems higher in tibialis anterior and diaphragm than in soleus...could the Authors speculate on possible muscle- or myofiber-type-specific regulation of miR-486?
- what was the rationale for focusing on MyoD and SRF as transcription factors regulating the ANK1 (and thus the miR-486) locus? Were other factors screened? Were these the only or most significant muscle-related factors found in the promoter region through sequence scanning? Also, with regards to Figure 1D, I don't think it helps in making the point that downregulation of MyoD and SRF with disease correlates with decreased miR-486 expression. Indeed, the heatmaps are quite heterogenous. I would suggest to simplify the panel and focus it on disease- and age-related decline in expression of MyoD, SRF and ANK1-5.
- it does not look like it was the case, but was there any difference in proliferation between miR-486-KO and -WT myoblasts?
- the heart data are quite striking, especially with regards to the overall dystrophic cardiomyopathy cadre (hypertrophy + dilation) and functional decline. However, the heart fibrosis data could be strengthened by either magnified insets of representative fibrotic scars, or images from Masson's trichrome, or more quantitative assays for fibrosis (for instance, hydroxyproline dosing or anything along those lines)
- as this is a microRNA-KO model, I'm curious about this: were there putative miR-486 binding sites (eCLIP peaks or computational predictions) in the top 10 upregulated genes in miR-486-KO versus -WT muscle as shown by RNA-seq? In other words, could the Authors speculate on whether the transcriptional changes (particularly the upregulated genes) are directly related to loss of the microRNA or rather linked to pathological remodeling? Another important question to that regard is: how much overlap was there between the RNA-seq "hits" and the eCLIP "hits" in terms of possible gene targets, i.e. putative miR-486 targets that were upregulated in mRNA levels in miR-486-KO versus WT? There was a fleeting comment in the Results section as "almost all targets were upregulated", but this should be quantitated and reported. After all, this is an important point of strength of the manuscript, i.e. the combination of a microRNA-KO with RNA-seq and eCLIP.
- (very minor) I think the abstract should be revised to increase clarity. There are several sentences that read oddly, including first, second and last.

Reviewer #2 (Comments to the Authors (Required)):

In their manuscript, Hightower et al generate mir-486 KO mice, which then they cross with mdx5cv mice. They show that mir-486 KO mice develop skeletal muscle and heart phenotypes and modify the phenotype of mdx5cv mice. RNA-seq and chimeric eCLIP were used to identify miR-486 in vivo targets. Bases on their results, the authors claim miR-486 is essential for normal muscle function, a driver of muscle remodeling in DMD, a useful biomarker for dystrophic disease progression. While interesting, the story presents several issues, and the authors claims are not fully supported by the data provided as indicated below.

Major points

1. Abstract, line 10: Given that miR-486 does not have RNA encoding for DGC components as direct targets, the claim that the authors have identified "direct drivers of dystrophinopathy pathologies" is not supported by the data provided and must be removed.

2. Page 6, line 69: It is not true that miR-486 "decrease in expression remained consistent throughout lifespan". Figure 1B shows that in soleus there is a "recovery" in miR-486 expression at 12 months, while in TA and diaphragm there is an age-dependent miR-486 decline.
3. Is dystrophin acute knockdown sufficient to downregulate miR-486?
4. In Figure S1D, the loading does not appear equal in all lanes and the signal is in any case too strong to appreciate any difference. The authors must repeat the western by loading lower, but equal amount of proteins in all lanes. Moreover, quantitative real-time PCR must be performed to further support no alteration in ANK1-5 level in mir-486 KO muscles.
5. A reduced fusion index is not sufficient to demonstrate a specific fusion defect. A result such that in Figure 1E-F can be caused by reduced proliferation, migration, fusion, or expression of the contractile apparatus. None of the experiments in the present manuscript or in Alexander et al 2014 demonstrate that miR-486 regulates specifically myoblast fusion. The authors must perform the required controls to demonstrate that miR-486 does not regulate myoblast proliferation, migration or expression of the contractile apparatus, or refrain from claims, which are not supported by the evidence provided.
6. Like the above point, the in vivo defects in mir-486 KO mice are not in line with a myoblast fusion defect, since mir-486 KO mice are able to form muscles and display regenerative capability.
7. It is not true that mir-486 KO mice show "increased fibrosis that was exacerbated in the dystrophic mdx5cv background. Figure 2D shows no altered fibrosis in mir-486 KO mice and no alteration of the levels of mdx5cv when combined with mir-486 KO. Figure S2A and S2C suggest that mir-486 KO is protective in the mdx5cv background. Similarly, Figure S3 shows several parameters that ameliorate in DKO mice compared to mdx5cv. For example, Figure S3B shows a complete rescue of cardiac fibrosis. It is tempting to speculate that miR-486 downregulation in DMD patients could be an adaptive, compensatory change.
8. The authors must provide the quantitative data from their chimeric eCLIP. It is not clear how many miR-486-specific associated mRNAs were identified by chimeric eCLIP, how many of those were lost in the mir-486 KO mice and what is the overlap between the miR-486-specific associated mRNAs and the transcripts identified by RNA-seq. What overlap with the transcripts downregulated or the transcript upregulated upon mir-486 KO identified by RNA-seq?
9. There is a disconnect between the results of Figure 4F and 5F. Given that RNA-seq and chimeric eCLIP identified genes belonging to very different categories, it appears that the alterations documented in Figure 4 are mostly secondary/indirect and not directly caused by mir-486 KO.
10. The results of Figure 5G are difficult to reconcile with the claimed role of miR-486 as a relevant player in DMD. Since mir-486 is downregulated in mdx mice, the expectation is to find similar behavior of miR-486 targets in mir-486 and mdx mice, while Figure 5G shows the opposite.
11. The authors must provide a detailed information about the statistical tests used to evaluate all the results reported in the manuscript.

Minor points

1. Abstract, line 4: develop must be corrected in development.
2. Abstract, line 5: since miRNAs function post-transcriptionally and on RNA molecules, it is not correct to write "gene target", which must be changed in RNA targets
3. Abstract, line 12: mir-486 must be in italic.
4. Introduction, line 40: The authors must provide recent references from the literature regarding ncRNAs controlling muscle regeneration.

Reviewer #3 (Comments to the Authors (Required)):

In the manuscript, Alexander and collaborators have furthered their previous studies on miR-486, an important factor in the regulation of DMD muscle pathology. miR-486, is included in myo-miR family even if does not have a muscle-specific expression: it is involved in important skeletal muscle development processes in fact targeting Pax7 it results to be highly up-regulated during muscle differentiation and, it was demonstrated that the down-regulation of miR-486 in normal myoblasts resulted in their impaired migration and differentiation due to direct modulation of Dock3/PTEN/AKT signaling by miR-486. Moreover, in a recent study, Alexander described reduced levels of miR-486 in dystrophic mice and the involvement of miR-486 has also been demonstrated in cardiac stress and cardiac syndromes.

In previous works by authors, it was elucidated that miR-486 lack alters the cell cycle kinetics of regenerated myofibers in vivo leading an impaired muscle regeneration and it was also revealed that miR-486 transgenic overexpression in dystrophic mice can ameliorate dystrophic pathologies, opening new investigations for therapeutic disease modulation.

Based on all these evidences, in the present work, the authors start new investigations on miR-486 for therapeutic DMD disease modulation. To deeply investigate miR-486 function in unperturbed muscle the authors used CRISPR technology and generated the mir-486 knockout mice (mir-486 KO). In Figure 2 and 3, analyzing the histology of miR-486 KO skeletal muscle and heart, the authors observed disrupted myofiber architecture, decreased myofiber size, decreased locomotor activity, increased cardiac fibrosis, and metabolic defects; interestingly, all these features were exacerbated when mir-486 KO mice were crossed to dystrophic mdx5cv mice.

Alexander and collaborators hypothesize that miR-486 could be a suitable DMD biomarker and aim to identify direct drivers of dystrophinopathy pathologies linked to miR-486 expression.

The authors choose an innovative way to identify direct miR-486 target straightly linked to muscle pathology overcoming the usual techniques that provide indirect methods to map miRNA targets (i.e., computational algorithms like TargetScan,...): they have adopted miR-eCLIP technology that enables precise mapping of direct miRNA-mRNA interactions transcriptome wide. Precise microRNA (miRNA) target identification is essential to understand post-transcriptional gene regulation and the miR-eCLIP is a recent technology that identifies at an extraordinary depth, direct in vivo miRNA targets using AGO2 immunoprecipitation, RNA-RNA ligation, and high-throughput sequencing as reported by authors in the scheme (Figure 5A). Conducting an integrated analysis based on RNA-sequencing and chimeric eCLIP sequencing the authors revealed 18 direct targets of miR-486 and they showed that many of the 18 targets found are already known as early stage DMD biomarkers including Ttn, Camk2a, Myom1, and Trim63.

Globally, authors demonstrated that in mice, the loss of mir-486 has an important impact on skeletal and cardiac muscle architecture, on fibrotic accumulation and both muscle and heart function and their findings highlight how directly or indirectly restoring the correct levels of miR-486 in dystrophic muscles could re-establish muscle homeostasis by improving DMD pathology paving the way for future studies.

Weakness and lack of manuscript:

In Figure 1 A the authors described a decreased expression of miR-486 in skeletal muscle biopsies from BMD similar to DMD ambulatory patients and they correlate this reduction to the loss or decrease of functional dystrophin. In Figure 1B they observed decreased expression of miR-486 in different skeletal muscles isolated from mdx mice at different ages and the reduction doesn't seem to be dependent by the severity of pathology being constant starting from 1 month of age to 12 months of age. This result seems to contrast what happens in the muscles of non-ambulatory DMD patients versus ambulatory DMD and BMD patients shown in figure 1A. An ImmunoHistoChemical analysis on muscle biopsies from DMD (different ages) and BMD patients could help to confirm this data. Did the authors speculate on this point?

I suggest to authors to include in the figure legend, the number of samples used for each experiment; this information is often missing.

In the experiments in Figure 1 E and 1F, the authors showed a decreased differentiation index for satellite cells isolated from miR-486 KO mice. In order to really understand that the results have to be interpreted as a defect in differentiation and not into reduced proliferation, the authors should include an experiment that goes in that direction, e.g. doing an edU pulse to satellite cells or detect a proliferation marker by immunofluorescence.

In Figure 4 the authors show their RNA-sequencing results on muscle RNA from miR-486 KO versus wild type mice. I think that the data generated could be strongly implemented if the authors consider also mdx muscle samples; overlapping results from KO to those of diseased dystrophic muscles will really help to draw a pathologic signature linked to miR-486. The proof of principle is the same used by authors in the next experiment of e-CLIP-miR.

Interesting, the RNA-sequencing analysis revealed extracellular matrix pathways in miR-486 KO muscle compared to WT mirroring what resulted in vivo histological analysis but looking at the tables in Figure 4G and H this result seems to get lost. I think it would be much more informative if the authors tabulated the top up-regulated and down regulated transcripts in line with the results of the pathway enrichment analysis.

For example in figure 4H among the top down-regulated genes it's clear a reduction in collagen1a1 expression, this is in contrast with pathway analysis and with the histological features described for miR486 KO mice that show a huge amount of fibrotic deposition. Did the authors look for the other collagen isoforms in particular the collagen 3a1? Could you please provide validations of genes most significant de-regulated by miR-486?

All the data referred to Figure 5 that represents the heart of the manuscript, needed to be validated not only by Real time PCR (as done). The authors could perform a classic Luciferase Activity assay to validate the functional interaction between miR-486 and the putative targets.

The results in Figure 5 can be completed adding the heatmap related to miR-486-KO vs mdx.

We would like to thank the reviewers for their excellent and thorough evaluation of our manuscript. We have made significant edits based on their comments. We apologize for delay in our response time as we encountered delays in experiments due to supply chain and COVID-related issues. The most notable edits to the manuscript involve the validation of the real time qPCR data in the *mir-486* KO mice, clarifications on the regulation of miR-486/ANK1-5 transcripts, and more insight into miR-486's function in isolated muscle cells. We will address all comments in a point-by-point fashion.

Reviewer 1:

1. "It is interesting to note that the magnitude of miR-486 downregulation seems higher in tibialis anterior and diaphragm than in soleus...could the Authors speculate on possible muscle- or myofiber-type-specific regulation of miR-486?" **Response:** We agree that there are likely fiber-type-specific regulation of miR-486 as recent snRNA-seq and single-myofiber sequencing papers have shown differential expression of miR-486 and the transcriptional regulators that drive its expression (Mizbani et al., *Development*, 2016; Petrani et al., *Nat. Comms.* 2020; McKeller et al., *Comms. Biol.*, 2021). Similarly in DMD, there is a long-reported preferential sparing of oxidative, slow-twitch myofibers that may explain our reported differential levels of miR-486 (Webster et al., *Cell*, 1988). We have added additional details in the Discussion.

2. "What was the rationale for focusing on MyoD and SRF as transcription factors regulating the ANK1 (and thus the miR-486) locus? Were other factors screened? Were these the only or most significant muscle-related factors found in the promoter region through sequence scanning? Also, with regards to Figure 1D, I don't think it helps in making the point that downregulation of MyoD and SRF with disease correlates with decreased miR-486 expression. Indeed, the heatmaps are quite heterogenous. I would suggest to simplify the panel and focus it on disease- and age-related decline in expression of MyoD, SRF and ANK1-5". **Response:** "We focused on the evolutionarily conserved myogenic transcription factor binding sites that flanked the miR-486 stem loop sequence and have previously shown to drive expression of key promoter elements of the miRNA (Small et al., *PNAS*, 2010), and those we performed ourselves via computational promoter analyses (rVISTA 2.0). Many of the additional myogenic factors (e.g. MRTF-A and MRTF-B) are validated transcriptional co-factors that bind to MyoD and SRF to regulate transcription (Cenik et al., *Development*, 2016). We have included a bit more rationale and scientific justification in the Results section to highlight this.

3. "It does not look like it was the case, but was there any difference in proliferation between miR-486-KO and -WT myoblasts?" **Response:** Similar to our previously published work showing lentiviral knockdown of miR-486 resulted in reduced myoblast proliferation (Alexander et al., *JCI*, 2014), we performed CLICK-iT proliferation studies of WT and miR-486 KO myoblasts and which showed a similar slight delayed proliferative capacity (Figures 1G and 1H).

4. "The heart data are quite striking, especially with regards to the overall dystrophic cardiomyopathy cadre (hypertrophy + dilation) and functional decline. However, the heart fibrosis data could be strengthened by either magnified insets of representative fibrotic scars, or images from Masson's

trichrome, or more quantitative assays for fibrosis (for instance, hydroxyproline dosing or anything along those lines)". **Response:** "We agree that the cardiac images previously presented did not fully capture the full fibrotic and histological analysis. We now provide enhanced magnification cardiac imaging pics to highlight the fibrotic lesions/areas, and quantification of the areas of fibrosis (Figures 3A).

5. "As this is a microRNA-KO model, I'm curious about this: were there putative miR-486 binding sites (eCLIP peaks or computational predictions) in the top 10 upregulated genes in miR-486-KO versus -WT muscle as shown by RNA-seq? In other words, could the Authors speculate on whether the transcriptional changes (particularly the upregulated genes) are directly related to loss of the microRNA or rather linked to pathological remodeling? Another important question to that regard is: how much overlap was there between the RNA-seq "hits" and the eCLIP "hits" in terms of possible gene targets, i.e. putative miR-486 targets that were upregulated in mRNA levels in miR-486-KO versus WT? There was a fleeting comment in the Results section as "almost all targets were upregulated", but this should be quantitated and reported. After all, this is an important point of strength of the manuscript, i.e. the combination of a microRNA-KO with RNA-seq and eCLIP." **Response:** Of our top 10 upregulated transcripts in the *mir-486* KO muscle (Figure 4G), we only observed *Ppp1r3c* as having a miR-486 binding site within its gene body. However, we would like to note that not all transcripts dysregulated were expected to be direct miR-486 targets as we later demonstrate that miR-486 controls many transcriptional pathways and that a significant amount of muscle remodeling occurs in the *mir-486* KO muscle.

We also agree with the reviewer that this area (Figure 5G; miR-486 direct binding targets) needed to be clarified and validated, as we know have included real time qPCR validation of the 18 miR-486 eCLIP-seq targets (Supplemental Figure 3). We also wish to highlight to the reviewers and readers that miRNA-mRNA binding does not always indicate inhibition as a growing aspect of post-transcriptional regulation via miRNAs is mRNA localization/splicing which has been confirmed by other miRNA-CLIP-seq experiments in tissues (La Rocca *et al.*, *Elife*, 2021; O'Connor *et al.*, *Genes Dev.*, 2021).

6. "(Very minor) I think the abstract should be revised to increase clarity. There are several sentences that read oddly, including first, second and last. **Response:** We have re-written and clarified these sentences and the entire Abstract.

Reviewer 2:

1. "Abstract, line 10: Given that miR-486 does not have RNA encoding for DGC components as direct targets, the claim that the authors have identified "direct drivers of dystrophinopathy pathologies" is not supported by the data provided and must be removed". **Response:** We agree with this point also raised by reviewer 1, and we have edited the text to reflect the role of these targets in *mir-486* KO muscle and as secondary contributors towards muscle pathologies in dystrophinopathy.

2. "Page 6, line 69: It is not true that miR-486 "decrease in expression remained consistent throughout lifespan". Figure 1B shows that in soleus there is a "recovery" in miR-486 expression at 12 months, while in TA and diaphragm there is an age-dependent miR-486 decline." **Response:** We have altered the text for this description as well as clarification as the samples are normalized to the WT control samples at the time of harvest, and not at the earlier (e.g. 1 month old) time point.

3. "Is dystrophin acute knockdown sufficient to downregulate miR-486?" **Response:** This is an interesting experiment as shRNAi of mouse *Dystrophin* in muscle cells has been previously shown by the Dickson lab to result in contractile apparatus and differentiation defects (Ghahramani Seno *et al.*, *BMC Genomics*, 2010). We examined their RNA array data (GSE20548 profile; Illumina 6.1 bead array) and while miR-486 itself was not on this array, several miR-486 transcriptional activators (*Srf*, *Mrtfa*,

Mrtfb; Figure 1D) and *Ank1* transcripts were all down-regulated (not shown). We speculate that a similar downregulation of miR-486 transcript also is occurring.

4. "In Figure S1D, the loading does not appear equal in all lanes and the signal is in any case too strong to appreciate any difference. The authors must repeat the western by loading lower, but equal amount of proteins in all lanes. Moreover, quantitative real-time PCR must be performed to further support no alteration in ANK1-5 level in mir-486 KO muscles." **Response:** We now have included updated western blot data taken at multiple development exposures (Sup. Figure 1D; below) showing no shift in ANK1-5 protein levels in the *mir-486* KO muscle lysates. Similarly, as requested our real time qPCR data demonstrated no significant change in mouse *Ank1-5* transcript in the *mir-486* KO muscle samples (Sup. Figure 1E; below).

5. "A reduced fusion index is not sufficient to demonstrate a specific fusion defect. A result such that in Figure 1E-F can be caused by reduced proliferation, migration, fusion, or expression of the contractile apparatus. None of the experiments in the present manuscript or in Alexander et al 2014 demonstrate that miR-486 regulates specifically myoblast fusion. The authors must perform the required controls to demonstrate that miR-486 does not regulate myoblast proliferation, migration or expression of the contractile apparatus, or refrain from claims, which are not supported by the evidence provided." **Response:** We agree with the authors that the terminology (e.g. *mir-486* KO muscle cell fusion defect) may not represent the most accurate description of the phenomena for which miR-486 deletion results in reduced differentiation levels. We have changed the terminology for this to reflect a "decreased differentiation capacity" as opposed to myogenic membrane-associated fusion (Figures 1E and 1F).

6. "Like the above point, the in vivo defects in *mir-486* KO mice are not in line with a myoblast fusion defect, since *mir-486* KO mice are able to form muscles and display regenerative capability." **Response:** We agree and have changed the terminology to reflect decreased myogenic differentiation capacity in *mir-486* KO muscle cells.

7. "It is not true that *mir-486* KO mice show "increased fibrosis that was exacerbated in the dystrophic *mdx5cv* background. Figure 2D shows no altered fibrosis in *mir-486* KO mice and no alteration of the levels of *mdx5cv* when combined with *mir-486* KO. Figure S2A and S2C suggest that *mir-486* KO is protective in the *mdx5cv* background. Similarly, Figure S3 shows several parameters that ameliorate in DKO mice compared to *mdx5cv*. For example, Figure S3B shows a complete rescue of cardiac fibrosis. It is tempting to speculate that miR-486 downregulation in DMD patients could be an adaptive, compensatory change." **Response:** We agree that the previous cardiac and skeletal muscle images were not fully reflective of the fibrotic changes, and we have included higher-resolution images and quantification of the fibrotic areas in the *mir-486* KO and *mir-486* KO:*mdx5cv* (dKO) mice (Figures 3A and 3B). We agree with the reviewer that there could be some compensation for miR-486 downregulation in DMD patients, and as we highlighted the loss of overall muscle myofibers that is replaced by fibrotic, inflammatory, and necrotic tissues in the DMD patients as they age.

8. "The authors must provide the quantitative data from their chimeric eCLIP. It is not clear how many miR-486-specific associated mRNAs were identified by chimeric eCLIP, how many of those were lost in

the mir-486 KO mice and what is the overlap between the miR-486-specific associated mRNAs and the transcripts identified by RNA-seq. What overlap with the transcripts downregulated or the transcript upregulated upon mir-486 KO identified by RNA-seq?" **Response:** We have now included the real time qPCR validation for all 18 miR-486 eCLIP target transcripts (Supplemental Figure 3).

9. "There is a disconnect between the results of Figure 4F and 5F. Given that RNA-seq and chimeric eCLIP identified genes belonging to very different categories, it appears that the alterations documented in Figure 4 are mostly secondary/indirect and not directly caused by mir-486 KO". **Response:** Many of the miR-486-regulated muscle gene pathways identified by the g:Profiler analysis did highlight muscle structural and extra-cellular matrix remodeling pathways (Figure 4F) for which we postulate are indirectly affected by miR-486-deficiency. Many, not all, of these pathways and transcripts are often dysregulated in DMD and other muscle disorders as we have highlighted. We also have cited the exact references towards these miR-486-regulated transcripts in muscle function and DMD disease throughout the results and discussion sections.

10. The results of Figure 5G are difficult to reconcile with the claimed role of miR-486 as a relevant player in DMD. Since mir-486 is downregulated in mdx mice, the expectation is to find similar behavior of miR-486 targets in mir-486 and mdx mice, while Figure 5G shows the opposite. **Response:** Similar to reviewer 1's earlier point, we stress that the miRNA-mRNA binding aspects does not always reflect inhibition of the mRNA transcript. Additionally, we do not claim that miR-486 directly is the sole cause for dystrophinopathy, but a contributing factor. We have also included our direct comparison of the miR-486 target transcripts between our WT vs *mir-486* KO muscle and WT vs *mdx*^{5cv} muscle RNA-sequencing experiments to reflect key differences in the regulation of transcript expression in two contexts (Figure 5G).

11. The authors must provide a detailed information about the statistical tests used to evaluate all the results reported in the manuscript." **Response:** We have added additional details to the statistical tests used in the Methods and Figure legends where appropriate.

Minor points

1. Abstract, line 4: develop must be corrected in development. **Response:** Corrected.
2. Abstract, line 5: since miRNAs function post-transcriptionally and on RNA molecules, it is not correct to write "gene target", which must be changed in RNA targets. **Response:** Corrected.
3. Abstract, line 12: mir-486 must be in italic. **Response:** Corrected.
4. Introduction, line 40: The authors must provide recent references from the literature regarding ncRNAs controlling muscle regeneration. **Response:** References have been added.

Reviewer 3:

1. "In Figure 1A the authors described a decreased expression of miR-486 in skeletal muscle biopsies from BMD similar to DMD ambulatory patients and they correlate this reduction to the loss or decrease of functional dystrophin. In Figure 1B they observed decreased expression of miR-486 in different skeletal muscles isolated from mdx mice at different ages and the reduction doesn't seem to be dependent by the severity of pathology being constant starting from 1 month of age to 12 months of age. This result seems to contrast what happens in the muscles of non-ambulatory DMD patients versus ambulatory DMD and BMD patients shown in figure 1A. An ImmunoHistoChemical analysis on

muscle biopsies from DMD (different ages) and BMD patients could help to confirm this data. Did the authors speculate on this point? I suggest to authors to include in the figure legend, the number of samples used for each experiment; this information is often missing.” **Response:** We have now clarified in the figure legend and methods that miR-486 transcript levels are normalized to the WT control miR-486 samples at each time point for the mouse. miR-486 levels in the human muscle biopsies are normalized to the healthy controls as well independent of age; although, they reflect biopsies of multiple similar ages and different *DYSTROPHIN* mutations. We apologize for the confusion with regards to the later mouse time points which might have led to mis-interpretation of our findings.

2. In the experiments in Figure 1 E and 1F, the authors showed a decreased differentiation index for satellite cells isolated from miR-486 KO mice. In order to really understand that the results have to be interpreted as a defect in differentiation and not into reduced proliferation, the authors should include an experiment that goes in that direction, e.g. doing an edU pulse to satellite cells or detect a proliferation marker by immunofluorescence. **Response:** Similar to reviewer 2’s point; we have both altered the language to reflect a decrease in overall myogenic differentiation capacity as opposed to myoblast fusion.

3. “In Figure 4 the authors show their RNA-sequencing results on muscle RNA from miR-486 KO versus wild type mice. I think that the data generated could be strongly implemented if the authors consider also mdx muscle samples; overlapping results from KO to those of diseased dystrophic muscles will really help to draw a pathologic signature linked to miR-486. The proof of principle is the same used by authors in the next experiment of e-CLIP-miR.” **Response:** We agree in principle that ideally performing chimeric-miR-486 eCLIP-seq in the mdx muscles might yield potential novel interactions between miR-486 and target transcripts. However, as we also stated above in response to reviewer 1, we believe that miR-486 exacerbates dystrophic pathologies by modulating the expression of key transcripts in DMD. We chose not to perform chimeric-miR-486-eCLIP-seq in the *mdx* muscles due to the overall low levels of miR-486 transcript detected in the *mdx* muscle samples.

4. “Interesting, the RNA-sequencing analysis revealed extracellular matrix pathways in miR-486 KO muscle compared to WT mirroring what resulted in vivo histological analysis but looking at the tables in Figure 4G and H this result seems to get lost. I think it would be much more informative if the authors tabulated the top up-regulated and down regulated transcripts in line with the results of the pathway enrichment analysis. For example in figure 4H among the top down-regulated genes it’s clear a reduction in collagen1a1 expression, this is in contrast with pathway analysis and with the histological features described for miR486 KO mice that show a huge amount of fibrotic deposition. Did the authors look for the other collagen isoforms in particular the collagen 3a1? Could you please provide validations of genes most significant de-regulated by miR-486? All the data referred to Figure 5 that represents the heart of the manuscript, needed to be validated not only by Real time PCR (as done). The authors could perform a classic Luciferase Activity assay to validate the functional interaction between miR-486 and the putative targets.” **Response:** Similar to the points raised by the other reviewers, we have now provided the qPCR validations for the 18 miR-486 target muscle transcripts (Supplemental Figure 3). However, we did not specifically detect *Col3a1* transcript as induced in expression in the *mir-486* KO muscles; although we detected several of the Collagen-6 transcripts (*Col6a2*, *Col6a3*) as induced. We have also included better images of the fibrotic lesions in the *mir-486* KO and dKO muscle samples, and provided a better explanation for the GO pathway analyses.

5. “The results in Figure 5 can be completed adding the heatmap related to miR-486-KO vs mdx.” **Response:** We have clarified the rationale for including the current heatmap comparing the transcript dysregulation in both conditions meant to provide information with regards to the miR-486 target transcripts in relevance to the condition and/or state tested (e.g. miR-486-deficiency versus dystrophin-deficiency). We do not believe that a dual comparison of *mir-486* KO versus *mdx*^{5cv} would be

informative as both conditions have either no or severely reduced miR-486 transcript expression and would not likely yield new miR-486-regulated target transcripts. We have also now included the real time qPCR validations (Supplemental Figure 3) for all 18 miR-486 transcript targets.

March 2, 2022

Re: Life Science Alliance manuscript #LSA-2021-01215-TR

Dr. Matthew Alexander
University of Alabama at Birmingham
Pediatrics
1918 University Blvd.
MCLM 922 #96
Birmingham, AL 35294

Dear Dr. Alexander,

Thank you for submitting your revised manuscript entitled "miR-486 is essential for muscle function and suppresses a dystrophic transcriptome" to Life Science Alliance. The manuscript has been seen by the original reviewers whose comments are appended below. While the reviewers continue to be overall positive about the work in terms of its suitability for Life Science Alliance, some important issues remain.

Our general policy is that papers are considered through only one revision cycle; however, we are open to one additional short round of revision. Please note that I will expect to make a final decision without additional reviewer input upon re-submission.

Please submit the final revision within one month, along with a letter that includes a point by point response to the remaining reviewer comments.

To upload the revised version of your manuscript, please log in to your account: <https://lsa.msubmit.net/cgi-bin/main.plex>
You will be guided to complete the submission of your revised manuscript and to fill in all necessary information.

B. MANUSCRIPT ORGANIZATION AND FORMATTING:

Sincerely,

Reviewer #1 (Comments to the Authors (Required)):

I think the Authors addressed most of the Reviewers' comments

Reviewer #2 (Comments to the Authors (Required)):

There are still important remaining points that have not been addressed satisfactorily by the authors, as listed below.

- For my question 3, it is unclear why the authors did not performed dystrophin knockdown themselves given it is a simple and fast experiment.
- There are a number of issues with the results described in Figures 2 and 3. First, the histology pictures do not seem at the same magnitude. Indeed, nuclear size is very different in the different pictures. Please provide pictures with the same magnification and resolution to exclude the possibility that non-representative area have been selected. For example, Figure 3B demonstrate that cardiac fibrosis is significantly reduced in dKO mice, hence the image of Figure 3A is misleading. Second, for most parameters analyzed the quantifications demonstrate that DKO mice have a phenotype comparable or less severe than single KO mice. The only parameter that is (maybe) significantly worsen in DKO is the percentage of central nuclei, and this could actually be seen as a rescue of regenerative capacity. In other words, the combination of the two defects (mir-486 KO and mdx5cv) in most of the cases produces just additive effects or even a rescue of disease. There is never a synergism significantly worsening the phenotype over the sum of mir-486 KO plus mdx5cv defects. Hence, mir-486 seems to function independently from mdx and the conclusion that "miR-486 is a key DMD biomarker whose downregulation drives pathogenic remodeling of dystrophic muscle" is not supported by the data provided.
- The answer to question 8 is unsatisfactory. How many miR-486-specific associated mRNAs were identified by chimeric eCLIP? How many of those were lost in the mir-486 KO mice? What is the overlap between the miR-486-specific associated mRNAs and the transcripts identified by RNA-seq? What overlap with the transcripts downregulated or the transcripts upregulated upon mir-486 KO identified by RNA-seq?

Reviewer #3 (Comments to the Authors (Required)):

Although the authors do not seem to have carried out the required experiments, by inserting new bibliography and editing the text they have managed to dispel my doubts and misgivings. The new pictures they have put up are also convincing.

We would like to thank the reviewers for their comments on the resubmitted version of our manuscript. We recognized that Reviewer 2 had additional questions with regard to the knockdown of dystrophin transcript and miR-486 expression regulation. We have addressed that experiment in addition to the additional target comparisons across the *mir-486* KO bulk RNA-seq, miR-486 chimeric eCLIP-seq, and *mdx*^{5cv} bulk RNA-seq cross dataset analyses in addition to more representative comparative histological images for each of the four cohorts. Here, we address all comments in a point-by-point fashion.

Reviewer 1:

Comment: I think the Authors addressed most of the Reviewers' comments. **Response:** We thank the reviewer for their helpful critique of our manuscript.

Reviewer 2:

Comment 1: For my question 3, it is unclear why the authors did not performed dystrophin knockdown themselves given it is a simple and fast experiment. **Response:** We have now performed the shRNAi knockdown experiment in mouse primary myotubes as recommended by the reviewer (**Supplementary Figure 4**).

Comment 2: There are a number of issues with the results described in Figures 2 and 3. First, the histology pictures do not seem at the same magnitude. Indeed, nuclear size is very different in the different pictures. Please provide pictures with the same magnification and resolution to exclude the possibility that non-representative area have been selected. For example, Figure 3B demonstrate that cardiac fibrosis is significantly reduced in dKO mice, hence the image of Figure 3A is misleading. **Response:** We have provided new images for **Figure 2A** (skeletal muscle histology) at standardized magnification of 20x across the four cohorts. Within each image set across the four cohorts in **Figure 3A** (cardiac chamber histology), magnification was standardized. We apologize for the confusion. We have highlighted areas of fibrosis and dilation across all 4 represented cohorts.

Comment 3: Second, for most parameters analyzed the quantifications demonstrate that DKO mice have a phenotype comparable or less severe than single KO mice. The only parameter that is (maybe) significantly worsen in DKO is the percentage of central nuclei, and this could actually be seen as a rescue of regenerative capacity. In other words, the combination of the two defects (mir-486 KO and *mdx*^{5cv}) in most of the cases produces just additive effects or even a rescue of disease. There is never a synergism significantly worsening the phenotype over the sum of mir-486 KO plus *mdx*^{5cv} defects. Hence, mir-486 seems to function independently from *mdx* and the conclusion that "miR-486 is a key DMD biomarker whose downregulation drives pathogenic remodeling of dystrophic muscle" is not supported by the data provided. **Response:** We acknowledge that while there is statistically less fibrotic areas in DKO compared to *mdx*^{5cv} as we have reported, there remains significant dystrophic pathologies that are as severe or slightly exacerbated as the *mdx*^{5cv} (dystrophin-deficient/DMD) mice as most of these skeletal muscle pathologies are already significantly fibrotic. We noted, as did the reviewer, the increased centralized myonuclei and smaller fiber sizes in the dKO mice as reflective of exacerbated impaired regeneration in the dKO mice (**Figures 2B and 2C**). Additionally, we detected comparable levels of overall fibrosis in the dKO mice to that of the *mdx*^{5cv} (**Figure 2D**). We postulate that the comprehensive phenotypic output as measured by gross dystrophic symptoms in skeletal muscle (e.g. overall fibrosis, myofiber size and centralized nucleation) is comparable or somewhat exacerbated in the dKO mice. The "recovery" as stated by the reviewer is not the decreased amount of fibrosis, but an exacerbation towards end-stage dystrophic pathologies caused by the loss of miR-486 expression. While the loss of *mir-486* did not significantly worsen the *mdx*^{5cv} pathologies in the double knockout (dKO) mice in all cardiac

functional output categories in direct comparison with *mdx*^{5cv} mice; the overall cardiac output, fractional shortening, and intraventricular septum thickness dystrophic dilated cardiomyopathy at similar or reduced levels to that of the *mdx*^{5cv} mice (**Figure 3C, 3E, 3J, and 3K**). As we state in the manuscript, we do not believe miR-486 alone is responsible for DMD-associated pathophysiologic effects, but rather does exacerbate them over time by modulating dystrophin-deficient factors via post-transcriptional regulation that negatively impact DMD disease outcomes. Additionally, our new data on AAV shRNAi knockdown of the large dystrophin muscle transcript (Dp427) in primary mouse myotubes demonstrates a strong link between miR-486 expression levels and the levels of dystrophin as shown (**Supplementary Figure 4**).

Comment 4: The answer to question 8 is unsatisfactory. How many miR-486-specific associated mRNAs were identified by chimeric eCLIP? How many of those were lost in the miR-486 KO mice? What is the overlap between the miR-486-specific associated mRNAs and the transcripts identified by RNA-seq? What overlap with the transcripts downregulated or the transcripts upregulated upon miR-486 KO identified by RNA-seq? **Response:** We apologize for any confusion in the initial reporting of our cross dataset analyses, and have now included two additional comparisons (**Supplementary Tables 2 and 3**) that explicitly list the overlapping number of miR-486 and DMD transcripts between the *mir-486* KO bulk RNA-seq, miR-486 chimeric eCLIP-seq, and *mdx*^{5cv} datasets. These are now listed as both a summary comparison of cross-platform datasets (**Supplementary Table 2**) and for each individual target that is dysregulated, as highlighted in **Supplementary Table 3**.

With regards to the number of “miR-486-specific” associated mRNAs identified by eCLIP, we identified 18 targets for which miR-486 binds to the target transcript to form a miRNA-mRNA hybrid when set at a significance cutoff ($p < 0.05$) compared to previous studies in cells using this platform (van Nostrand et al., Nat. Meth. 2016). With regards to the question on the number of “lost” transcripts between the *mir-486* KO bulk RNA-seq versus miR-486 chimeric eCLIP-seq, we first want to highlight to the reviewer that the chimeric eCLIP-seq does not assume that miRNA:mRNA binding results in knockdown/decrease in expression as compared with transcriptional expression types of changes normally detected in bulk RNA-sequencing. Thus, none of the 18 chimeric eCLIP-seq targets identified from the chimeric eCLIP-seq (as defined $p < 0.05$) were also strongly dysregulated in the *mir-486* KO versus WT TA muscle dataset; however, 16 of the 18 targets identified by miR-486 chimeric eCLIP-seq were increased in expression in the *mir-486* KO mice and 14 of the 18 miR-486 chimeric eCLIP-seq targets were increased in the *mdx*^{5cv} (DMD) mouse muscles (**Supplementary Table 2**). We have also listed the exact overlapping targets in **Supplementary Table 3**. Again, we emphasize here (and in the manuscript) that the miR-486 targets identified via chimeric eCLIP-seq are a snapshot of miR-486:mRNA target binding at a given time point (e.g. adult WT male TA muscle) and not necessarily representative of a significant miRNA-mediated transcript knockdown/inhibition at that particular given time point. We fully acknowledge to the reviewer and the reader that additional chimeric eCLIP-seq during a time period where miR-486 may be more abundant in expression (e.g. embryonic myogenesis), additional miR-486 targets of statistical significance (meeting the $p < 0.05$ threshold) could be identified using this platform. We have modified the language in the text to reflect the need to integrate multiple platforms and perform individual target validation for all sequencing results.

Reviewer 3:

Comment: Although the authors do not seem to have carried out the required experiments, by inserting new bibliography and editing the text they have managed to dispel my doubts and misgivings. The new pictures they have put up are also convincing. **Response:** We thank the reviewer for their helpful critique of our manuscript and have put up additional comparisons between the *mir-486* KO bulk RNA-seq, miR-486 chimeric eCLIP-seq, and *mdx*^{5cv} datasets as also suggested by Reviewer 2 (**Supplementary Tables 2 and 3**).

April 19, 2022

RE: Life Science Alliance Manuscript #LSA-2021-01215-TRR

Dr. Matthew Alexander
University of Alabama at Birmingham
Pediatrics
1918 University Blvd.
MCLM 922 #96
Birmingham, AL 35294

Dear Dr. Alexander,

Thank you for submitting your revised manuscript entitled "miR-486 is essential for muscle function and suppresses a dystrophic transcriptome". We would be happy to publish your paper in Life Science Alliance pending final revisions necessary to meet our formatting guidelines.

- please note that Supplementary references should be incorporated into the main manuscript
- please add your main, supplementary figure, and table legends to the main manuscript text after the references section; all figure legends should only appear in the main manuscript file
- please consult our manuscript preparation guidelines <https://www.life-science-alliance.org/manuscript-prep> and make sure your manuscript sections are in the correct order
- please upload your Table in editable .doc or excel format
- we encourage you to revise the figure legends for figures 1 and 3 such that the figure panels are introduced in an alphabetical order
- please add callouts for Figure 1G-H to your main manuscript text
- supplementary methods section is uploaded separately. Please incorporate it into the main M&M section in the manuscript text

A. FINAL FILES:

B. MANUSCRIPT ORGANIZATION AND FORMATTING:

Sincerely,

April 20, 2022

RE: Life Science Alliance Manuscript #LSA-2021-01215-TRRR

Dr. Matthew Alexander
University of Alabama at Birmingham
Pediatrics
1918 University Blvd.
MCLM 922 #96
Birmingham, AL 35294

Dear Dr. Alexander,

Thank you for submitting your Research Article entitled "miR-486 is essential for muscle function and suppresses a dystrophic transcriptome". It is a pleasure to let you know that your manuscript is now accepted for publication in Life Science Alliance. Congratulations on this interesting work.

DISTRIBUTION OF MATERIALS:

Again, congratulations on a very nice paper. I hope you found the review process to be constructive and are pleased with how the manuscript was handled editorially. We look forward to future exciting submissions from your lab.

Sincerely,
